# Climate response to off-equatorial stratospheric sulfur injections in three Earth System Models - Part 1: experimental protocols and surface changes

Daniele Visioni[1], Ewa M. Bednarz[1], Walker R. Lee[1], Ben Kravitz[2,3], Andy Jones[4], Jim M. Haywood[4,5], and Douglas G. MacMartin[1]

[1]Sibley School of Mechanical and Aerospace Engineering, Cornell University, Ithaca, NY, USA
[2]Department of Earth and Atmospheric Science, Indiana University, Bloomington, IN, USA
[3]Atmospheric Sciences and Global Change Division, Pacific Northwest National Laboratory, Richland, WA, USA
[4]Met Office Hadley Centre, Exeter, UK
[5]College of Engineering, Mathematics and Physical Sciences, University of Exeter, Exeter, UK

**Correspondence:** Daniele Visioni (dv224@cornell.edu)

**Abstract.** There is now a substantial literature of climate model studies of equatorial or tropical stratospheric $SO_2$ injections that aim to counteract the surface warming produced by rising concentrations of greenhouse gases. Here we present the results from the first systematic intercomparison of climate responses in three Earth System Models where the injection of $SO_2$ occurs at different latitudes in the lower stratosphere: CESM2-WACCM6, UKESM1.0 and GISS-E2.1-G. The first two use a modal aerosol microphysics scheme, while two versions of GISS-E2.1-G use a bulk aerosol (One-Moment Aerosol, OMA) and a two-moment (Multiconfiguration Aerosol TRacker of mIXing state, MATRIX) microphysics approach, respectively. Our aim in this work is to determine commonalities and differences between the climate model responses in terms of the distribution of the optically reflective sulfate aerosols produced from the oxidation of $SO_2$, and in terms of the surface response to the resulting reduction in solar radiation. A focus on understanding the contribution of characteristics of models transport alongside their microphysical and chemical schemes, and on evaluating the resulting stratospheric responses in different models is given in the companion paper (Bednarz et al., 2022). The goal of this exercise is not to evaluate these single point injection simulations as stand-alone proposed strategies to counteract global warming; instead we determine sources and areas of agreement and uncertainty in the simulated responses and, ultimately, the possibility of designing a comprehensive intervention strategy capable of managing multiple simultaneous climate goals through the combination of different injection locations.

We find large disagreements between GISS-E2.1-G and the CESM2-WACCM6 and UKESM1.0 models regarding the magnitude of cooling per unit of aerosol optical depth (AOD) produced, from 4.7 K per unit of AOD in CESM2-WACCM6 to 16.7 K in the GISS-E2.1-G version with two-moment aerosol microphysics. By normalizing the results with the global mean response in each of the models, and thus assuming that the amount of $SO_2$ injected is a free parameter that can be managed independently, we highlight some commonalities in the overall distributions of the aerosols, in the inter-hemispheric surface temperature response and in shifts to the Inter-Tropical Convergence Zone, and also some areas of disagreement, such as the extent of the aerosol confinement in the equatorial region and the efficiency of the transport to polar latitudes. In conclusion, we

demonstrate that it is possible to use these simulations to produce more comprehensive injection strategies in multiple climate models. However, large differences in the injection magnitudes can be expected, potentially increasing inter-model spreads in some stratospheric quantities (such as aerosol distribution) while reducing the spread in the surface response in terms of temperature and precipitation; furthermore, the selection of the injection locations may be dependant on the models' specific stratospheric transport.

## 1 Introduction

A climate model is an imperfect representation of the real climate system, having to deal with simplified processes across spatial and temporal scales, bound both by technical limits to the computational capabilities and by fundamental uncertainties in some of the underlying processes. Reducing such uncertainties can be a complicated process and may leverage various strategies: i) by simulating current and past climates for which we have available observations, the skills of current climate models can be evaluated and the reliability of future projections assessed (Brunner et al., 2020). ii) The use of large ensembles of simulations can further help to narrow down some of the uncertainties by isolating biases due to internal model variability over time (Deser et al., 2012; Lehner et al., 2020). iii) Finally, the use of multiple climate models can help to clarify some of the sources of uncertainty by analyzing the response to an external forcing in multiple independent models. This has been done in numerous generations of the Climate Model Intercomparison Project (CMIP) (Masson and Knutti, 2011; Boucher et al., 2013; Tebaldi et al., 2021), which addresses the effects of an increased $CO_2$ forcing on the global and regional surface climate (Hawkins and Sutton, 2011). Other modeling exercises such as the Chemistry-Climate Model Initiative (Morgenstern et al., 2018) have used models with explicit atmospheric chemistry coupling (which many of the CMIP models lack) to determine the future evolution of key atmospheric variables, such as stratospheric ozone (Dhomse et al., 2018) and stratospheric circulation (Eichinger et al., 2019). Finally, model intercomparisons have been designed to constrain the response of climate models to explosive volcanic eruptions, trying to understand both the evolution of the aerosol cloud over time (Timmreck et al., 2018) and its climatic effects (Zanchettin et al., 2016; Clyne et al., 2021; Zanchettin et al., 2022).

The assessment of simulated climate intervention techniques presents all of the challenges discussed above, but also novel ones. The proposed injection of sulfur dioxide ($SO_2$) to temporarily cool the planet (stratospheric aerosol intervention, SAI) while greenhouse gas levels are reduced (Crutzen, 2006) has been studied with climate models in the last two decades, and multiple intercomparisons have been carried out spearheaded by the Geoengineering Model Intercomparison Project (GeoMIP, Kravitz et al. (2011)). The first SAI experiments, GeoMIP G3 and G4 (Kravitz et al., 2011), prescribed the injection of $SO_2$ in the equatorial stratosphere (in a time-varying or constant manner, respectively). However, the modeling teams were free to specify the exact details of the injection "in the same manner as they simulate a volcanic eruption"; this led to differences in the approaches chosen. For instance, the University of L'Aquila Chemistry-Climate Model (ULAQ-CCM) injected $SO_2$ between 18 and 25 km of altitude using a Gaussian distribution, whereas the Goddard Chemistry-Climate Model (GEOSCCM) injected $SO_2$ uniformly between 16 and 25 km of altitude. The varying experimental protocols resulted in large differences

in the simulated aerosol distributions, which compounded the differences coming from the differences in the representation of aerosol microphysics amongst the models (Visioni et al., 2017). More recently, the GeoMIP experiment G6 (Kravitz et al., 2015) prescribed variable injections of $SO_2$ between 10°N and 10°S, and between 18 and 20 km of altitude. However, some models lacking interactive aerosol treatment prescribed an aerosol distribution from sources not entirely consistent with the original prescription for models without interactive microphysics, resulting in challenges in the attribution of the causes of some of the resulting discrepancies (Visioni et al., 2021). Lastly, the test-bed experiment described by Weisenstein et al. (2021) prescribed two injection scenarios (one uniformly between 18 and 20 km, and between 30°N and 30°S; and one in two precise locations, 30°N and 30°S, at 19 km of altitude) to understand inter-model differences between injections of $SO_2$ and direct injections of accumulation mode $H_2SO_4$ to better control the size of the resulting aerosols.

In parallel, simulations with the Community Earth System Model version 1 with the Whole Atmosphere Community Climate Model (CESM1(WACCM)) were carried out examining how changes in the location of the sulfate injection matters for the climatic impacts. Tilmes et al. (2017) injected $SO_2$ at 4 locations (30°N, 15°N, 15°S and 30°S) and showed that they resulted in different aerosol distributions and in different atmospheric and surface responses (see also Richter et al. (2017); Tilmes et al. (2018b)). Further, MacMartin et al. (2017) showed that these $SO_2$ injection locations could be combined to achieve multiple climate goals with the use of a feedback algorithm capable of deciding each year how much $SO_2$ to inject at each of the aforementioned latitudes (Kravitz et al., 2017). The four locations of injection were necessary to control the overall geographical distribution of the simulated aerosols and, thus, to constrain not only the global mean surface temperatures, but also to make sure that the large-scale inter-hemispheric surface temperature gradient and the equator-to-pole surface temperature gradient are maintained. This approach was adopted in the Geoengineering Large Ensemble experiment (GLENS, Tilmes et al. (2018a)) that studied the feasibility and climate impacts of a potential SAI deployment strategy. By differentiating the injection locations, numerous side-effects and adverse impacts of SAI could be reduced compared to an equatorial injection strategy (Kravitz et al., 2019). Up to now, all of the above assessments, and the exploration of other strategies, have been performed with only one climate model - CESM1(WACCM).

In this work we aim to perform the first systematic intercomparison of the stratospheric and surface climate responses to the injection of $SO_2$ at different latitudes in the stratosphere. We use three comprehensive Earth System Models (ESMs) with interactive sulfate aerosol treatment and a set of carefully designed experiments with a single point of injection per experiment. Our goals are: i) to robustly evaluate similarities and differences in the simulated sulfate aerosol distributions and in the resulting surface and atmospheric responses; ii) to elucidate the areas and sources of intermodel differences, in particular the roles of the specific model microphysical schemes and the biases in climatological transport; iii) to determine the reliability of estimates of the surface responses to SAI previously performed in MacMartin et al. (2017) and the middle atmospheric responses performed in Tilmes et al. (2018b); Richter et al. (2017) using the multi-model approach; and iv) to lay the ground for an intermodel comparison of SAI simulations achieving the same set of surface temperature goals defined in Kravitz et al. (2017) and mentioned above using a feedback algorithm. In this paper we discuss the simulated aerosol fields and

their effects on zonal mean surface temperatures and precipitation; we then use that information to determine the magnitude of $SO_2$ injection needed in each location to obtain the desired temperature targets. In the companion paper (Bednarz et al. (2022), hereafter PART2), the simulated differences in the aerosol distribution are explained in terms of the model differences in atmospheric circulation, and the resulting SAI impacts on stratospheric temperatures, chemistry and circulation are discussed.

## 2 Earth System models used

We performed our simulations with three Earth System Models (ESMs): CESM2-WACCM6, UKESM1.0 and GISS-E2.1-G; for the latter, two different versions with two different aerosol treatments were used. One simulation for each model has been performed. All models have full atmosphere-ocean-land coupling as well as an explicit aerosol treatment and stratospheric chemistry. A brief description is provided below for all of the models, and a discussion of the differences in the aerosol treatments of particular importance for this work is given at the end of this section.

### 2.1 CESM2-WACCM6

The Community Earth System Model, version 2, with the Whole Atmosphere Community Climate Model version 6 (CESM2-WACCM6, Gettelman et al. (2019); Danabasoglu et al. (2020), hereafter CESM2) is used with a comprehensive stratospheric and upper atmospheric chemistry and interactive aerosol microphysics using the Modal Aerosol Module (MAM4) (Liu et al., 2016). Its horizontal resolution is 1.25° longitude by 0.9° latitude, with 70 vertical levels and a model top at about 140 km. An evaluation of the model response to past volcanic eruptions was performed in Mills et al. (2017) and Schmidt et al. (2018) using the previous version of the model, CESM1(WACCM). Unlike the CESM2 version described in Danabasoglu et al. (2020), which includes both comprehensive tropospheric and stratospheric chemistry, the version used here includes comprehensive stratospheric chemistry but only a simplified chemistry of importance in the troposphere; this is thus similar to the chemistry scheme in the CESM1(WACCM) version used for the previous geoengineering studies, e.g. Tilmes et al. (2018a).

### 2.2 GISS-E2.1-G

NASA's ModelE is developed by the Goddard Institute for Space Studies (GISS). The current version, ModelE2.1, has a horizontal resolution of 2.5° longitude by 2° latitude and 40 vertical layers extending up through the mesosphere (model top of ~80 km). The version used in this study, GISS-E2.1-G, has a fully interactive ocean with 32 vertical layers, based on the Russell ocean (Kelley et al., 2020). While GISS ModelE also has the ability to use the HYCOM ocean (Kelley et al., 2020), that option is not employed here. GISS has two methods of representing aerosol microphysics that we employed here. The bulk aerosol treatment (referred to as One-Moment Aerosol, OMA in Bauer et al. (2020), called 'GISS-OMA' in this study) involves specification of an aerosol dry radius, and the aerosols grow hygroscopically as a function of the relative humidity (Koch et al., 2006). The more complex aerosol treatment (referred to as Multiconfiguration Aerosol TRacker of mIXing state, MATRIX in Bauer et al. (2008), called 'GISS-MATRIX' in this study) involves computation using quadrature of moments and can represent aerosol microphysical growth via condensation and coagulation. While not strictly a modal treatment, as it

computes aerosol size using a quadrature of moments approach (usually termed a two-moment approach), it is most comparable to the modal treatments used in the other two models. For anthropogenic, tropospheric aerosols, a comparison between the two aerosol treatments in GISS is presented in Bauer et al. (2020). Both aerosol methods have a representations of heterogeneous halogen chemistry on the aerosol surfaces and are coupled to the same chemistry scheme, which in the stratosphere includes chlorine and bromine chemistry and a treatment of polar stratospheric clouds (Shindell et al., 2006).

## 2.3 UKESM1.0

The UKESM1 Earth system model (Sellar et al., 2019) was developed jointly by the UK's Met Office and Natural Environment Research Council and consists of the physical atmosphere-land-ocean-sea-ice model HadGEM3-GC3.1 (Kuhlbrodt et al., 2018) coupled to components which deal with terrestrial and ocean biogeochemistry (Wiltshire et al., 2021; Yool et al., 2021; Walters et al., 2019). Its comprehensive stratospheric-tropospheric chemistry (StratTrop) scheme is described in Archibald et al. (2020) and includes detailed tropospheric chemistry and a somewhat simplified stratospheric chemistry with 'lumped' treatment of long-lived halogenated ozone depleting substances. The model is further coupled to a modal aerosol microphysics scheme described in Mulcahy et al. (2018) and Mann et al. (2010) (GLOMAP-mode). The horizontal resolution is 1.875° longitude × 1.75° latitude, with 85 vertical levels up to ∼84 km on terrain-following hybrid height coordinate. An evaluation of simulations of past volcanic eruptions has been recently performed in Dhomse et al. (2020) for on older version of this model using a similar version of the aerosol scheme, as well as in Aubry et al. (2021).

## 2.4 Differences between the aerosol microphysics schemes used

None of the models used in this study employ a sectional aerosol microphysics, where the size domain is divided into intervals, or bins, and the evolution of the number concentrations in each size bin is calculated separately for each species. Instead, these models employ a more simplified approach, either modal (or comparable to modal in the case of GISS-MATRIX), where the aerosol population is described by a number of log-normal distributions, or bulk, where the size distribution is prescribed; we will describe both briefly below.

CESM2 and UKESM1 use a very similar modal approach, where the aerosol population is described by at least three main modes (at least for sulfate) called Nucleation (only for UKESM1), Aitken, Accumulation and Coarse, whose distribution is assumed to be lognormal with a fixed geometric standard deviation $\sigma_g$ and a geometric mean diameter that can vary between a certain predefined size range: particle number and mass are transferred to the larger mode when the diameter exceeds the upper limit for that mode. In each mode, all aerosol species are considered internally mixed, that is, they are described by a single size distribution; this has been shown to lead to changes in upper tropospheric aerosol concentrations when large quantities of stratospheric sulfate settle down, unrealistically reducing the size distribution and thus the settling velocities of species that shouldn't interact with the sulfate (Visioni et al., 2022).

The GISS in its bulk version incorporates a much simpler aerosol parametrization, where the size distribution is specified for each aerosol species. There is no calculated number concentration and water uptake effects are solely dependent on relative humidity for gravitational settling purposes. GISS-MATRIX sits in between the complexity of the other two models and GISS-OMA: the sulfate population is described by two lognormal distributions (Aitken and Accumulation) of fixed $\sigma_g$, but separated from other aerosol species, defined as particles with a diameter smaller or larger than 0.1 $\mu$m respectively. Instead of tracking the size distribution itself, however, only key moments of the aerosol population such as number and mass are tracked, based on the quadrature method of moments framework described in McGraw (1997), leading to a set of equations to be solved for each population described in detail in Bauer et al. (2008). To prevent the mean diameter of the Aitken mode from approaching that of the Accumulation mode too often, a transfer function that completely moves all particles from the Aitken to the Accumulation mode when the two sizes approach in value is used (see section 2.8 in Bauer et al. (2008)). For a straightforward comparison, all information for the aerosol populations in each model is described in Table 1. For GISS-OMA, the only prescribed value is the bulk diameter (which for simplicity we included in the Accumulation row in the table) with no assumptions over the geometric standard deviation; the actual value of 0.3 $\mu$m used in the current version of GISS used here is different from the one given in Koch et al. (2006) (0.4 $\mu$m).

We note that in the GISS with bulk treatment, there is one specified aerosol dry radius for all sulfate aerosols, both tropospheric and stratospheric. We used the default aerosol size, which is calibrated to represent tropospheric sulfate and the background sulfate layer in the stratosphere but is far too small for the aerosols that would result from a high stratospheric loading of sulfate. Past experiments using an earlier version of this model increased the aerosol size, resulting in a better match to stratospheric sulfate aerosols compared to observations but was far larger than should have occurred for tropospheric aerosols. This had non-negligible effects on radiative forcing, tropospheric chemistry, and aerosol deposition (e.g., Kravitz et al., 2009; Pitari et al., 2014), limiting the ability to compare the SAI run with a corresponding baseline run. The approach chosen in the present study avoids these issues, but in doing so limits the applicability of the GISS-OMA simluations to SAI. Nevertheless, these simulations serve as a useful point of comparison and reveal understanding.

As shown in Table 1, the modal aerosol schemes present differences in the definition of the size ranges per each mode, and we will show that this significantly change in which modes the aerosol find themselves in. While the width of the smallest mode is similar between all three models, it is significantly different for the coarse mode, being 1.2 for CESM2 and 2 for UKESM. Considering that $\sigma_g$ is a parameter that remains constant in the whole atmosphere, this difference may have arisen if the two models considered different types of aerosols size distributions for which they desired their mode to be more similar to. For instance, if one consider the stratospheric size distribution of aerosols post-Pinatubo, $\sigma_g$ has values lower than 1.5 (Post et al., 1997), whereas for background conditions or small eruption, the value is closer to 1.8 (Wurl et al., 2010; Doeringer et al., 2012). As detailed in Mills et al. (2017), CESM did modify its coarse mode in MAM4 to better reproduce the Pinatubo plume.

**Table 1.** Values for the size range (diameter, D, in $\mu$m) and geometric standard deviation $\sigma_g$ for sulfate of each of the modes (when applicable) for the four participating models. Information collected from Koch et al. (2006) for GISS-OMA, Bauer et al. (2008) for GISS-MATRIX, Walters et al. (2019) for UKESM and Liu et al. (2012) with integrations from Mills et al. (2016) for CESM2. All diameter are considered here as dry, as models calculate water uptake afterward depending on local humidity.

| Mode name | CESM2 size ($\mu$m) | CESM2 $\sigma_g$ | GISS-OMA size ($\mu$m) | GISS-OMA $\sigma_g$ | GISS-MATRIX size ($\mu$m) | GISS-MATRIX $\sigma_g$ | UKESM1 size ($\mu$m) | UKESM1 $\sigma_g$ |
|---|---|---|---|---|---|---|---|---|
| Nucleation | - | - | - | - | - | - | D<0.01 | 1.59 |
| Aitken | 0.015<D<0.053 | 1.6 | - | - | D<0.1 | 1.6 | 0.01<D<0.1 | 1.59 |
| Accumulation | 0.058<D<0.48 | 1.6 | 0.3 (fixed) | NA | D>0.1 | 1.8 | 0.1<D<0.5 | 1.40 |
| Coarse | D>0.4 | 1.2 | - | - | - | - | D>0.5 | 2 |

## 3   Experimental protocol

### 3.1   Baseline emission scenario

For these experiments, we selected the baseline Shared Socioeconomic Pathway (SSP) 2-4.5 (Meinshausen et al., 2020). Each experiment is run for 10 years starting on January $1^{st}$ 2035. The choice of a particular background scenario is of secondary importance here, as all comparisons of the SAI responses will be performed relative to the control SSP2-4.5 simulation, thereby evaluating climate changes due to solely the increased sulfate aerosol concentrations. However, in planning for future, more comprehensive experiments (as those detailed in Richter et al. (2022) and MacMartin et al. (2022)) we use a similar background greenhouse gas scenario.

### 3.2   Specifics for the SO$_2$ injection

In the experiments discussed in Tilmes et al. (2017), the altitudes of SO$_2$ injections were defined in terms of a fixed height above the annual mean tropopause. This led to two different injections altitudes, one for 15°N and 15°S injections at 25.0 km and one for injections at 30°N and 30°S at 22.4 km. Here we define a single injection altitude for all latitudes at 22 km. We chose this injection altitude for two separate reasons. First, it is closer to the upper limit of altitudes achievable by traditional aircraft (Smith, 2020) while still being sufficiently far above the tropopause for aerosols not to be removed too quickly (as opposed to 25 km, which is likely too high for practical considerations). Second, we aimed to inject SO$_2$ in all models at the same latitude but also in just one gridbox; however, there is a challenge in prescribing a fixed injection altitude in kilometers but having a hybrid height vertical coordinate (UKESM1) or a hybrid sigma-pressure coordinate (CESM2, GISS). After performing some tests, the specific choice of 22 km ensured that all three models could always be reasonably expected to inject in the same gridbox, as that altitude is not close to the vertical edges of a gridbox in any of the models. In CESM2, this meant injecting at the gridbox bounded by the 47 hPa and 39 hPa pressure interfaces (with a midpoint of 43 hPa), which has an average geometric height of 21.6 km at all considered latitudes of injection and is 1.2 km thick. In UKESM1, this meant injecting at the level with

a midpoint height of 21.8 km, which is roughly 1 km thick. In GISS, this meant injecting at the gridbox bounded by the 43 and 31 hPa interfaces, with an approximate midpoint of 37 hPa (23.1 km assuming a 7 km scale height) and approximately 2.3 km thick.

$SO_2$ is injected continuously throughout the year in the same quantity in each experiment: each simulation includes the injection in only one location, either 30°N, 15°N, 0°N, 15°S or 30°S, and at one longitude (180°E for GISS and CESM2, 0°E for UKESM1) at the altitude described above. Simulations are performed for 10 years each. The selected $SO_2$ injection rate in each model is 12 Tg-$SO_2$ per year, to allow for an easier comparison with past simulations with CESM1-WACCM in Tilmes et al. (2017).

**4   Results**

**4.1   Changes in stratospheric aerosols**

Fig. 1 shows the evolution of the global mean changes in stratospheric Aerosol Optical Depth (AOD, calculated for all models at 550 nm) and surface temperatures. A relatively good agreement is found between the CESM2, GISS-MATRIX and UKESM1 in terms of the global mean AOD responses, whereas the global mean temperature response is similar for CESM
and UKESM1, and for the two GISS versions separately. GISS-OMA presents AOD values larger than the other two models and GISS-MATRIX, with differences ranging from values that are two times larger (for equatorial and 15°N injections) to 33% larger (for 30°N injections). Larger values of sulfate burden can result in a number of non-linear effects (Niemeier and Timmreck, 2015), for instance on atmospheric dynamics and thus the latitudinal aerosol distribution (Visioni et al., 2020) or on surface climate (Simpson et al., 2019; Jiang et al., 2019). The uncertainties in the SAI process that leads from $SO_2$ injection
to surface impacts can be separated into three different parts: i) at each latitude how much $SO_2$ is needed to achieve a certain optical depth (i.e. the efficiency of $SO_2$ to $H_2SO_4$ conversion and of the removal processes); ii) the resulting distribution of AOD under a specific injection location(s) (largely driven by large scale dynamics and mixing) and iii) the impacts of a specific aerosol distribution on climate. Simulations with fixed $SO_2$ injections at fixed locations, as used in this manuscript, allow to explore better points (i) and (ii). In contrast, simulations like those described in (Kravitz et al., 2017), where the amount of
$SO_2$ injected is adjusted each year in order to achieve some specified surface temperature goals, allow to better understand the response to some specific pattern of AOD forcing (as the injection rate can be adjusted to achieve a desired AOD). In the following analyses we separate and discuss both the overall simulated zonal mean response and that normalized by the global magnitude of the response to highlight these different contributors to the overall uncertainty.

Figure 2 shows the latitudinal distributions of the simulated stratospheric AOD. The inter-model spread is different depending on the injection location, both in terms of overall magnitude and in terms of the spatial distribution. Equatorial injections show by far the greatest differences, with UKESM1 simulating twice as much AOD in the tropics compared to CESM2, and the two GISS versions differing by up to a factor of 5 between each other. Similar differences between UKESM1 and CESM2 have

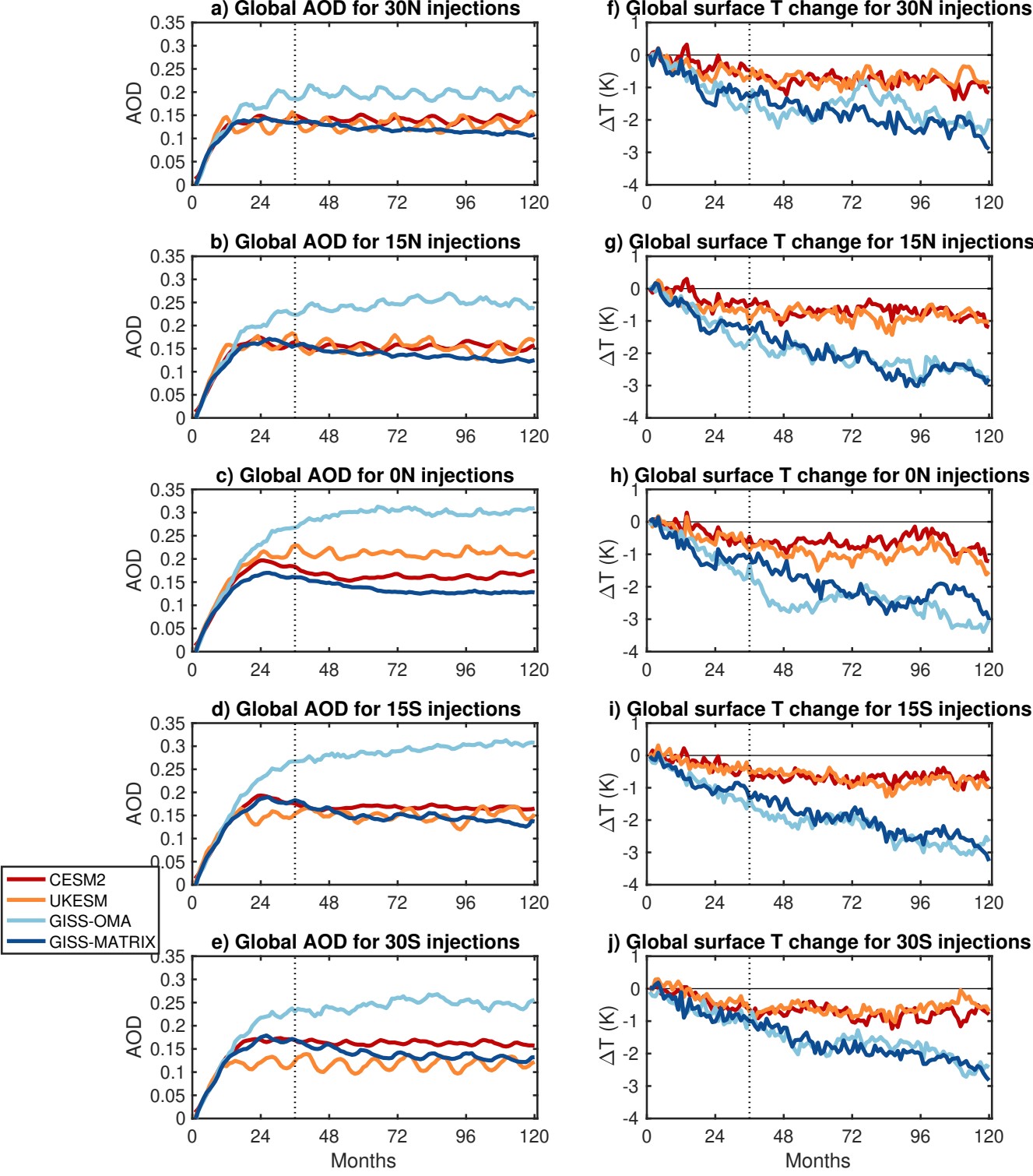

**Figure 1.** Timeseries of global mean monthly mean changes in stratospheric AOD (a-e) and surface temperature (f-f) resulting from the single-point injections at 30°N (a, f), 15°N (b, g), 0°N (c, h), 15°S (d, i) and 30°S (e, j) compared to the corresponding baseline SSP2-4.5 simulation in each of the models. The dashed line in all panels represent the month after which averages are performed for following figures (i.e. the last 7 years of simulations).

also been documented for the GeoMIP G6 experiment in Jones et al. (2021). Part of these differences are driven by differences in the global mean values of AOD: as mentioned above, AOD in GISS-OMA is over two times larger than in GISS-MATRIX for the equatorial injection case. In the right panels of Figure 2 the zonal mean AOD values are scaled by the respective global mean values; this highlights the differences in the latitudinal distribution of the responses. This way, the difference in the normalised magnitudes of the equatorial peak (defined here as the average between 5°N and 5°S) between CESM2 and UKESM1 is reduced from 1.8 to 1.4; and similarly the two GISS versions show much more similar results, especially in the tropical region. In general, injections in the Southern Hemisphere (SH) show larger inter-model differences compared to the Northern Hemisphere (NH) injections, especially at high latitudes. In the 30°S case, global mean values are more than twice as large in GISS-OMA (0.25) compared to UKESM1 (0.12), and the SH high-latitudinal AOD differs by a factor of three. In the NH, the largest difference is 60% between UKESM1 (0.12) and GISS-OMA (0.20) in the 30°N case, and high-latitudinal AOD differs by a factor of two. When normalised, the differences are largely reduced, but a clear differentiation remains for UKESM1 which shows a larger tropical confinement of aerosols than the other models. There are also some notable differences in the interannual variability between models: while both GISS versions and CESM2 have standard deviations that range between 1 and 5% of the mean values (depending on the precise latitude), UKESM1 shows a much larger variability (up to 33 % in the 30°N injection case at 60°N-90°N). In future longer simulations this observation would need to be considered when considering also the interannual variability of the surface changes.

The differences in AOD can be better understood by looking at the changes in aerosol mass mixing ratios (Figure 3) and the overall sulfate column burden (Figure 4). Compared to the other models, UKESM1 shows a much stronger confinement of the aerosol mass in the tropical pipe (for equatorial injections, the peak is 1.5 $\mu$g-$SO_4$/kg-air, whereas it is 1.1 $\mu$g-$SO_4$/kg-air for CESM2 and GISS-OMA, and 0.5 for GISS-MATRIX), and far less transport in the opposite hemisphere for both 15° injections. On the other hand, CESM2 shows a much stronger poleward transport of sulfate aerosols, as indicated by aerosol values twice as large for 30° injections compared to other models considering values above 60° in latitude. All of these characteristics can largely be explained by the characteristics of the climatological transport in each of these models, as discussed in depth in PART2.

GISS-OMA also shows a larger poleward transport in all cases, but the underlying cause for this might be different between GISS and CESM2. As analyzed in depth in PART2, the baseline stratospheric dynamics is very similar between the two GISS realizations and, hence, the differences in the overall aerosol distribution can't be found there. Analyses of the stratospheric aerosol surface area density (SAD, shown in PART2 due to its importance for stratospheric ozone chemistry) indicates a far larger number of particles in GISS compared to the other models. The SAD in GISS-OMA is three times as large poleward of 60° compared to CESM2, even for mass values that only differ by 30% (for instance, in the 30°N injection case). GISS-OMA simulates much smaller particles (the prescribed bulk dry radius is 0.15 $\mu$m) that imply higher residence times, as gravitational settling is much lower (Visioni et al., 2022), and so the particles are more easily transported towards the pole even in the presence of a similar dynamical regime as compared with the other GISS realization. The presence of aerosols at higher

altitudes even close to the poles (where the large-scale circulation should be transporting aerosols downward) further supports this observation. Smaller aerosols are also more efficient scatterers, explaining why the optical depth differences are larger than the mass differences.

Finally, panels f-h in Figure 4 give an overview for all cases of the ratio between the overall mass of the produced aerosols (shown alone in panel 4f) and the injected amount of $SO_2$. This ratio, shown in panel 4g, represents the lifetime of the added sulfate: as we are in a steady state, where no new mass is added to the global burden, this lifetime can be calculated as the burden divided by its constant source (the injection) (Visioni et al., 2018). All models show larger amounts of mass for injections closer to the equator, due to the tropical confinement increasing the aerosols lifetime. This effect is less noticeable for GISS-OMA, which shows far less confinement. However, models show a distinct difference in the ratio between overall mass of $SO_4$ and resulting AOD (panel 4h): this indicates substantial inter-model differences in the average size of the aerosol, with obvious consequences for the efficiency of the cooling that we will analyze next.

A comparison of the effective radius ($R_{eff}$) is shown in Figure 5. The values for $R_{eff}$ are indirectly derived for all models as $R_{eff}$ is not a direct output for either UKESM1 and GISS. Therefore, we use the common available output of mass mixing ratio ($\chi$) and number concentration ($N$) for each mode in each model to derive the mean radius $r_i$. We calculate the mean volume $v_i$ using

$$v_i = \frac{\chi_i}{\rho_{sulfate} * N_i} \tag{1}$$

where $\rho_{sulfate}$ is the density of sulfate as considered in each model in kg/m$^3$ (usually 1770 kg/m$^3$ as in Liu et al. (2012)). The mean radius is then derived considering that the aerosol population is assumed to be a lognormal distribution with a geometric standard deviation $\sigma_g$, and is thus connected to the mean volume by

$$v_i = \frac{4}{3} * \pi * r_i^3 * exp(\frac{9}{2} * ln^2(\sigma_g)) \tag{2}$$

Finally, $R_{eff}$ is calculated considering the definition

$$R_{eff} = \frac{\sum_i r_i^3 N_i}{\sum_i r_i^2 N_i} \tag{3}$$

where i is the number of modes considered, so 3 for CESM2 (all modes relevant to sulfate in MAM4, Liu et al. (2016)), 4 for UKESM1 and 2 for GISS-MATRIX. The single wet radii for each mode are shown in the Supplementary Material in Fig. S1. The validity of our derivation is ensured by the comparison of the derived values of $R_{eff}$ in CESM2 and the ones obtained as a direct output from the CESM2 simulations (not shown); while the off-line derivation leads to slightly smaller $R_{eff}$ values than calculated online in the model, the overall results are similar enough for a confident comparison. A number of important

features become apparent: the population-weighted radius in GISS-MATRIX is almost always larger than the GISS-OMA one, confirming our observation over the differences in high latitudinal concentration. CESM2 also presents a radius always larger than the GISS-MATRIX and UKESM1 one, which explains most of the discrepancies between AOD and column mass between the CESM2 and GISS models: even if CESM2 shows larger mass concentrations, smaller particles such as those in GISS are more efficient scatterers, thus GISS AOD is more comparable to CESM2. If dynamics were similar between the two models, one would also expect less mass in CESM2 compared to GISS due to reduced lifetime caused by the increased gravitational settling, but in this case, the dynamical differences dominate in determining the final mass concentrations; these differences are discussed more in depth in PART2 and indicate a substantially stronger climatological shallow branch of the Brewer-Dobson circulation in CESM2 compared to other models (see Figure 2 in PART2). UKESM1 has intermediate radii between CESM2 and GISS-MATRIX.

Looking at the mean radii for each mode (Figure S1), differences in the microphysical approaches become more evident between the models, also based on the values given in Table 1. Given that UKESM1 defines the Coarse mode as particles larger than 0.5 $\mu$m, all of the stratospheric aerosols are found in the Accumulation mode (as they are in GISS), whereas CESM2, that defines the Coarse mode threshold at around 0.4 $\mu$m, simulates most of the sulfate aerosols in that mode, and very few in the Accumulation mode. This particular observation shouldn't significantly influence the aerosols' behavior in the stratosphere, since the coagulation and condensation processes happen in all modes. It could, however, have some impacts on the tropospheric interactions of sulfate with cloud nuclei; for instance some models (such as CESM2) treat Coarse mode and Accumulation mode aerosols differently when calculating the presence of heterogeneous and homogeneous ice nuclei (see for example Visioni et al. (2022) for a discussion of undesired side effects this might have on ice nucleation rates in the upper troposphere in CESM1(WACCM)).

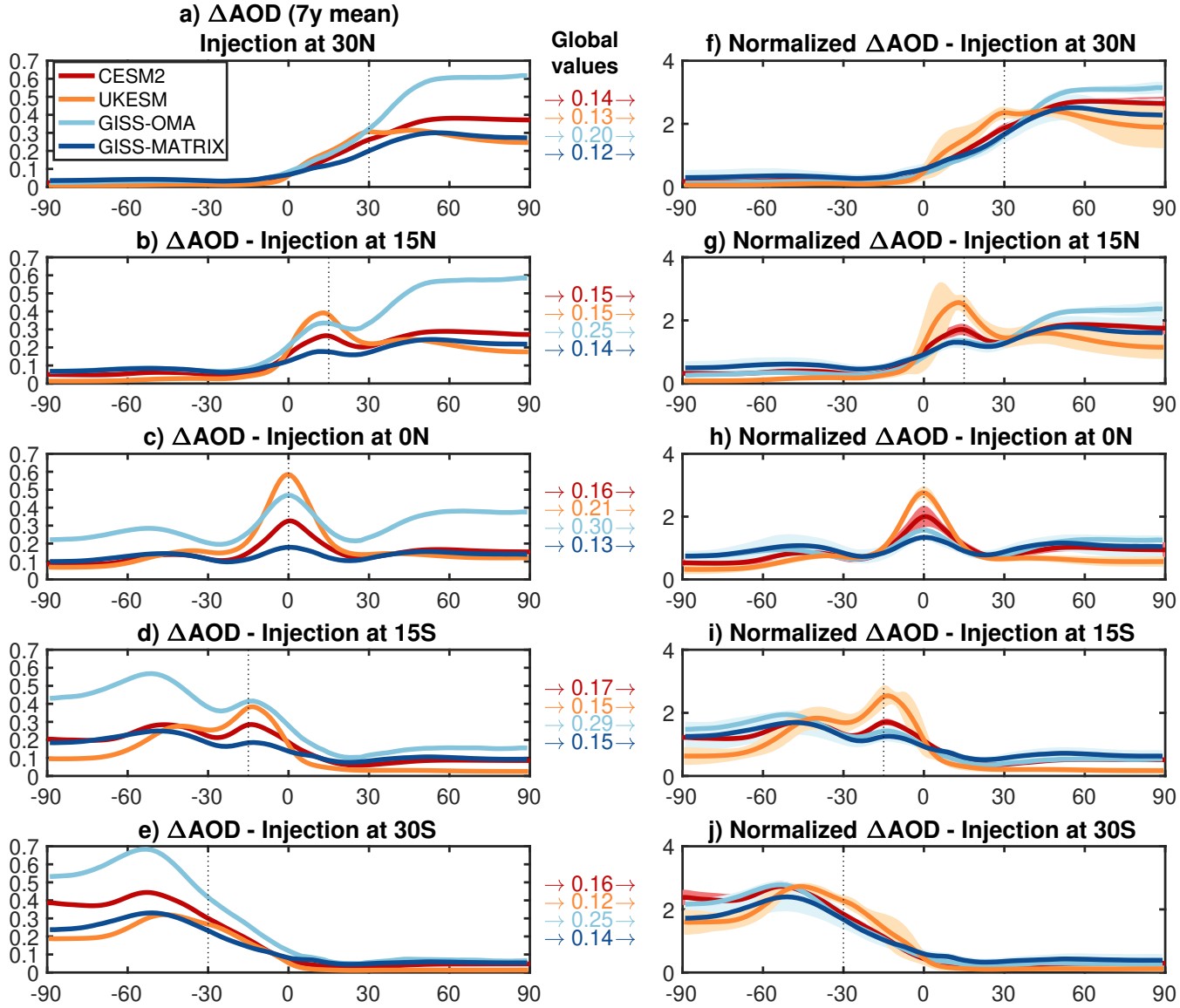

**Figure 2.** (a-e) Zonal and annual mean increase in stratospheric AOD resulting from single-point injections at 30°N (a), 15°N (b), 0°N (c), 15°S (d) and 30°S (e), averaged over the last seven years of simulation. (f-j) Zonal mean increase normalized to the global value of AOD in each simulation in the respective experiment; shading of the same color represent the internal variability (1 $\sigma$ standard deviation) over the same period of time. For each experiment, the global mean value used in the normalization is reported between the panels on the left and right. Dashed vertical lines indicate the locations of the SO$_2$ injections.

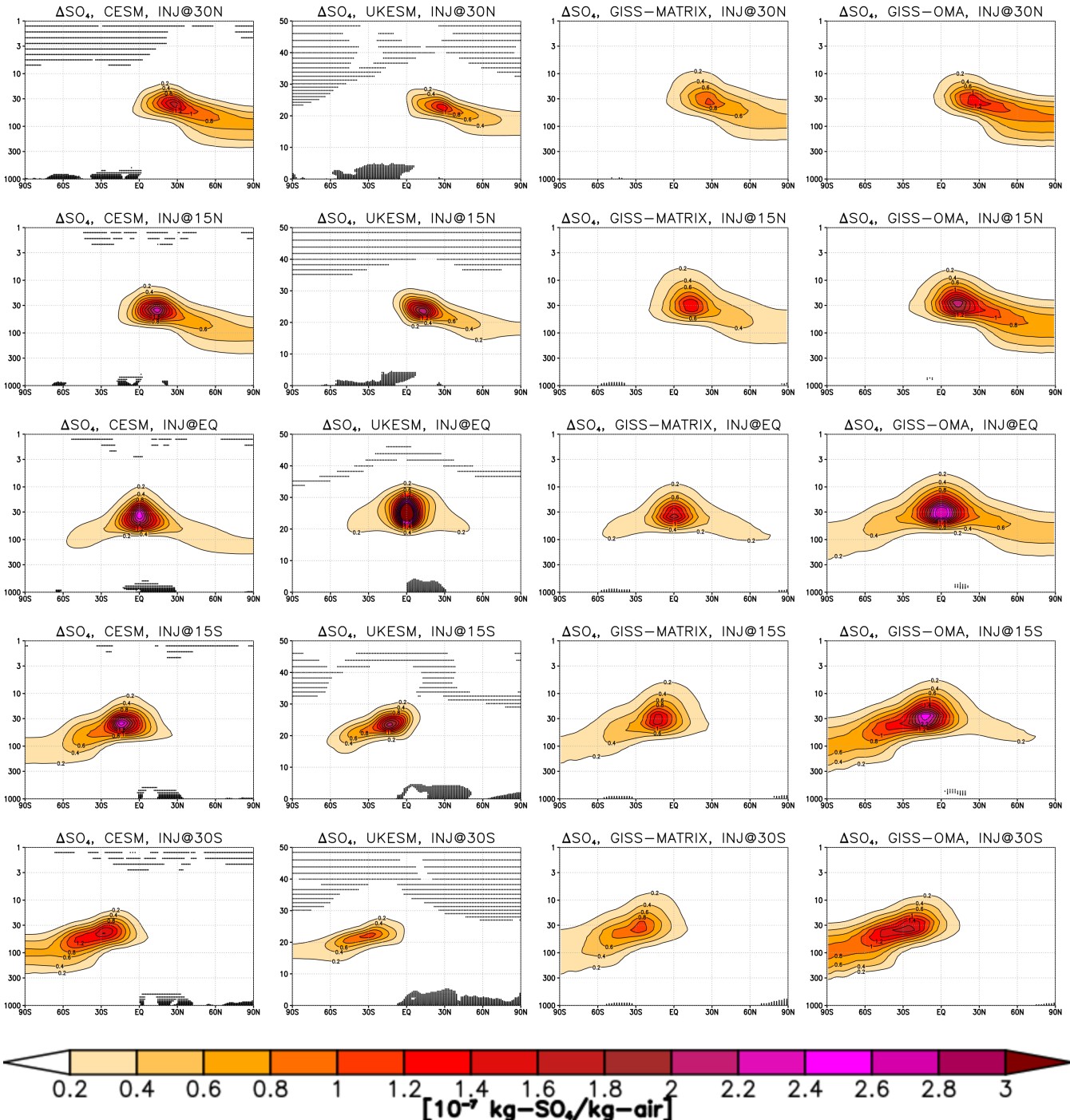

**Figure 3.** Zonal and annual mean increase in SO$_4$ mass mixing ratio (in $10^{-7}$ kg-SO$_4$/kg-air) for CESM2 (first column), UKESM1 (second column), GISS-MATRIX (third column) and GISS-OMA (fourth column) and all injection locations, 30°S (first row), 15°S (second row), 0°N (third row), 15°N (fourth row) and 30°N (fifth row). Hatched areas indicate locations where changes are not statistically significant (using a two-sided t-test at 95% confidence level).

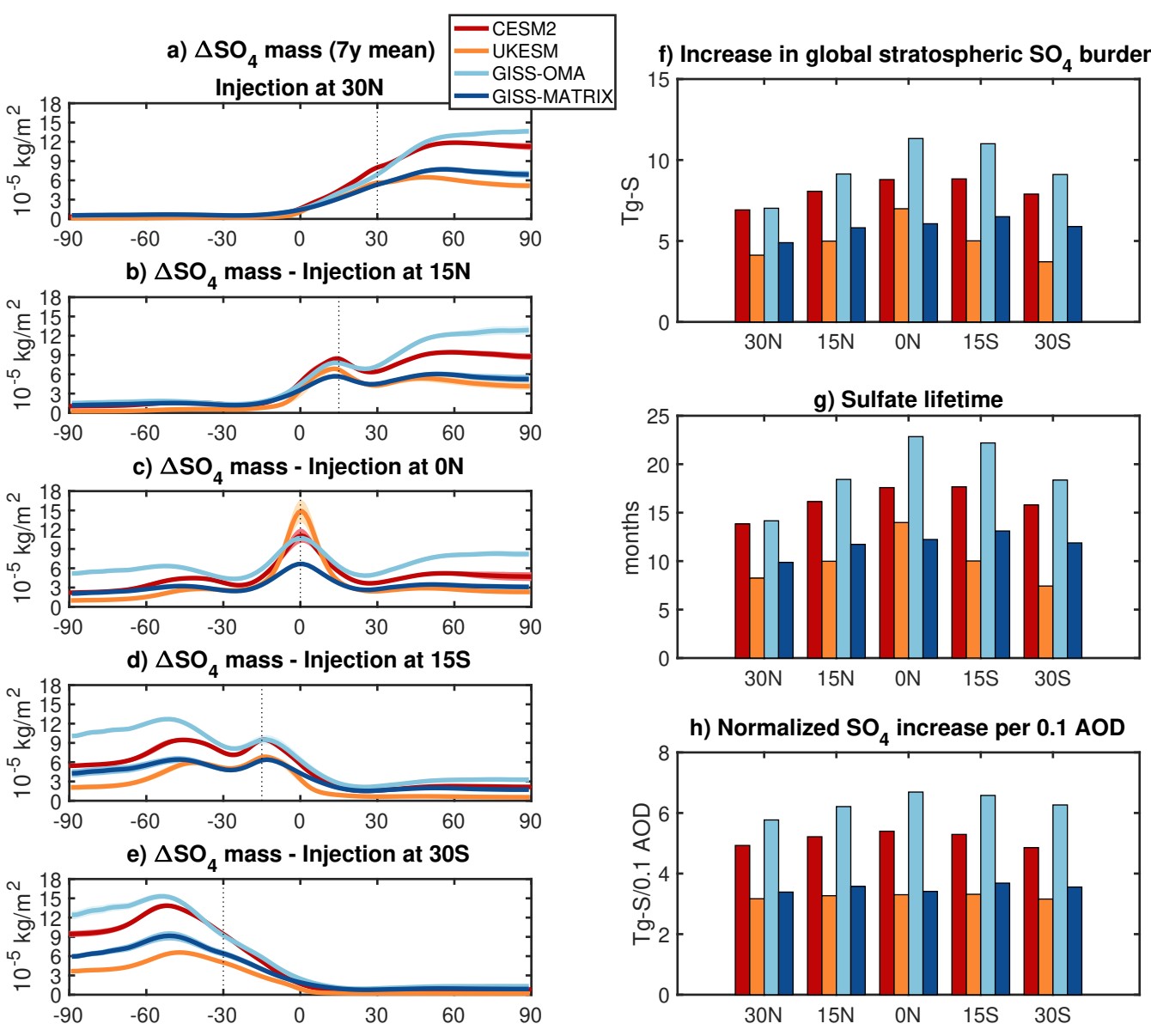

**Figure 4.** (a-e) Zonal and annual mean increase in total $SO_4$ column burden for each experiment, $30°N$ (first row), $15°N$ (second row), $0°N$ (third row), $15°S$ (fourth row) and $30°S$ (fifth row). f) Total global increase in stratospheric sulfate burden (in Tg-S). g) Stratospheric sulfate lifetime (in months). h) Global increase in column sulfate burden, normalized by the obtained global mean increase in stratospheric AOD.

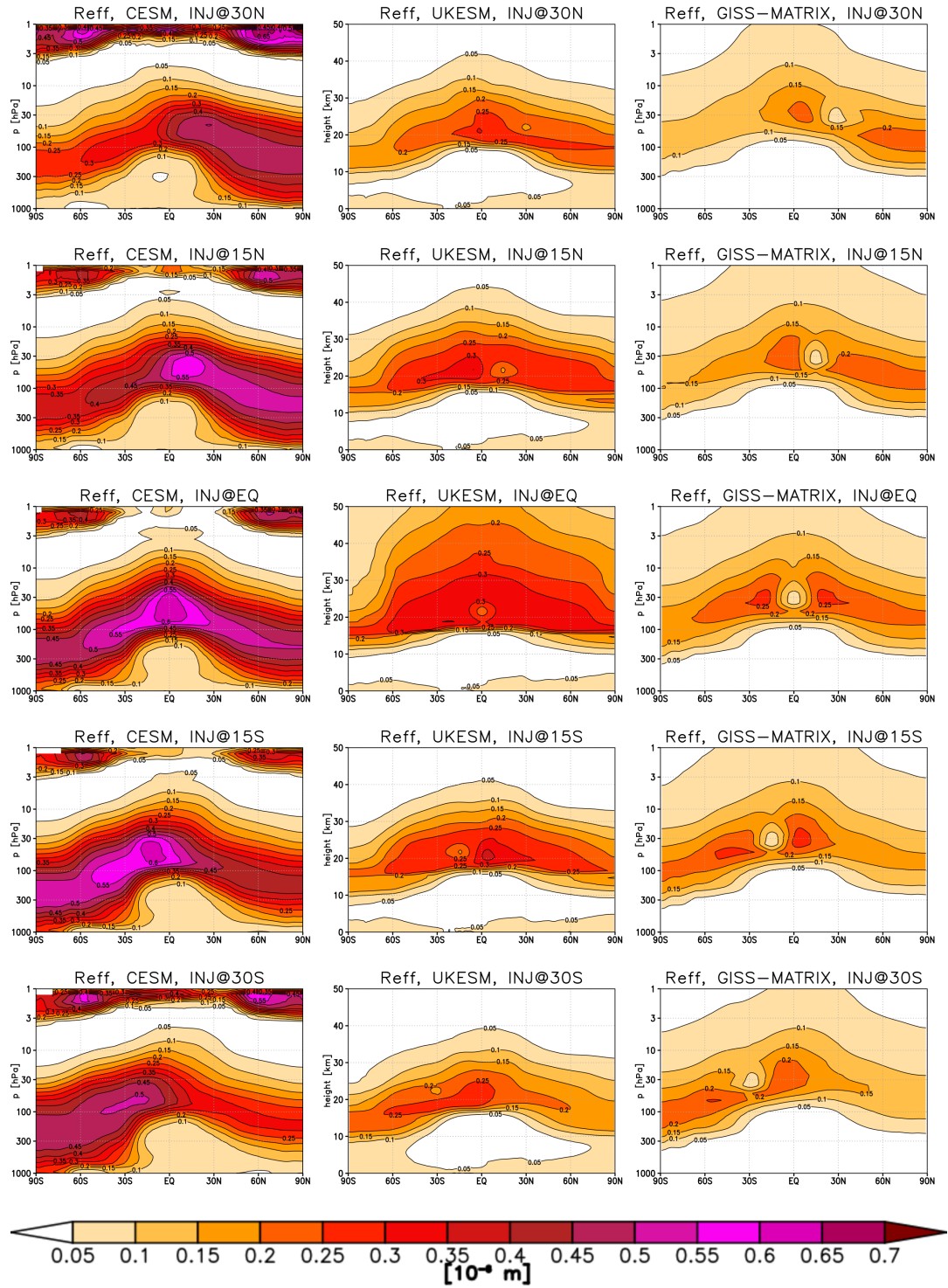

**Figure 5.** Effective radius (in $10^{-6}$ m) in CESM2 (left), UKESM1 (center) and GISS-MATRIX (right) simulations for each experiment, 30°S (first row), 15°S (second row), 0°N (third row), 15°N (fourth row) and 30°N (fifth row).

The resulting zonal mean cooling achieved by the AOD increase in each simulation is shown in Figure 6, including both the absolute temperature changes and values normalized by the global means. Differences between models are still evident, but a direct correlation with the AOD changes in Fig. 2 is not present; for instance, the general temperature responses between the two GISS realizations are much more similar than their simulated AOD, as was already visible in Fig. 1. This is because temperature responses compound many sources of uncertainty, including not only those related to the magnitude and spatial pattern of the simulated stratospheric AOD (thus related to the model microphysics and transport), but also to the resulting radiative, dynamical and chemical processes operating in the troposphere and lower stratosphere (including the associated changes in ozone and water vapour, which are discussed in PART2) that modulate how much cooling AOD produces and how the regional patterns of cooling differ. We find very large differences between the models in the magnitude of the global mean cooling per unit AOD (Figure 7). CESM2 and UKESM1 show, on average, 4.7 K and 5.4 K cooling per unit of AOD, respectively. This is roughly in line with the multi-model average of GeoMIP G6sulfur experiment (six models, resulting in 4.6 K per unit AOD for injections between 10°N and 10°S in GeoMIP (Visioni et al., 2021), compared to the 3.9 K and 4.8 K cooling per unit AOD in CESM2 and UKESM1, respectively, for the equatorial injections here).

In contrast, the GISS models show significantly larger cooling per unit AOD, with 8.6 K on average in GISS-OMA and 16.7 K on average in GISS-MATRIX; the latter is more than three times as high as in CESM2. As noted for Fig. 1, this large normalized difference is mainly due to differences in the global cooling produced between the two GISS realizations (the sensitivity to the aerosols amount); even if there is a much smaller amount of AOD simulated in GISS-MATRIX, it results in a similar level of surface cooling as is simulated in GISS-OMA under much larger AOD values. The cause for the discrepancies between the two GISS realizations is most likely found in similar tuning procedures for the model (described for GISS-E2.1 in Kelley et al. (2020)); while these mainly target the global radiative balance, they often also try to constrain the aerosol forcing (this is described for instance in Schmidt et al. (2017), where both GISS and CESM1 tuning procedures are explained in some detail). The two GISS versions used here show large differences in the baseline aerosol optical depth from sulfate: in the mid-latitudinal NH, where anthropogenic sulfate aerosol production is maximized, GISS-MATRIX shows on average a 0.03 AOD, just like CESM2, while GISS-OMA shows on average a 0.11 AOD (see Figure S2). Further investigations will be warranted in the baseline GISS simulations to understand if this difference is relevant for the observed differences in the sensitivity to stratospheric aerosols between the two versions.

Another possible contributor to the differences in the magnitude of global cooling simulated in GISS could be the substantial tropospheric ozone reduction simulated in the two model versions (See Section 3.3.4 of PART2). As a greenhouse gas, reduction in tropospheric ozone would thus act to magnify the cooling from the direct aerosol forcing, and this effect is generally not present in CESM2 or UKESM1.

The normalized values in the right panels of Figure 6 show some level of consistency in the temperature response amongst the models. In particular, the models agree that equatorial injections tend to cool the NH more than the SH, and that the cooling

produced by the NH injections is strongest in the NH (this is particularly marked in CESM2 for the 15°N injection), whereas the cooling produced by the SH injections is more evenly distributed between the two hemispheres. In contrast, the models tend to disagree with respect to responses in the NH high latitudes much more than for the SH high latitudes; this can be explained by the role of dynamical atmospheric variability and sea-ice variability, which in the NH both may contribute to increasing models' spread (Banerjee et al., 2021).

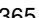

**Figure 6.** (a-e) Zonal and annual mean change in surface temperatures (K) resulting from the single-point injections at either 30°N (a), 15°N (b), 0°N (c), 15°S (d) or 30°S (e), averaged over the last seven years of simulation. (f-j) Zonal mean changes normalized with the corresponding global mean values in each experiment; shading of the same color represent the interannual variability (1 $\sigma$) over the same period of time. For each experiment, the global mean value used in the normalization is reported between the panels on the left and right.

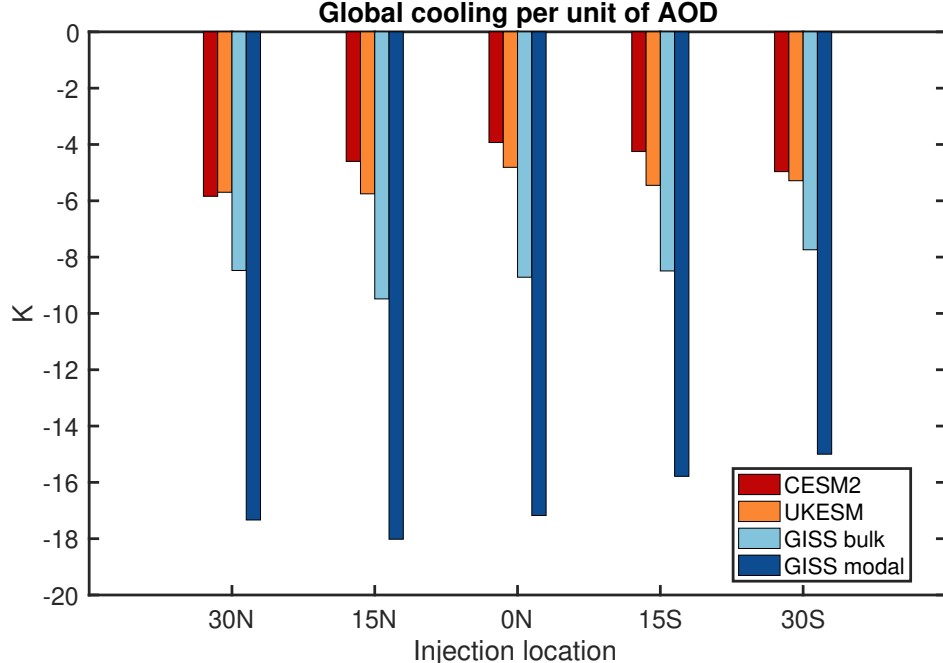

**Figure 7.** Global mean cooling achieved in each experiment and injection location per unit increase in the associated global mean stratospheric AOD.

Finally, in Figure 8 we show the zonal mean changes in precipitation. In this case, we show both the magnitude of the change in mm/day, and the corresponding % changes with respect to the baseline precipitation pattern in each model. The precipitation response also depends on the climatological precipitation, which differs more between the models than the climatological temperature does (not shown). We find a general qualitative agreement between the models in the tropics. All models show shifts in tropical precipitation depending on the hemisphere of injection, with decreased precipitation in that hemisphere and increased precipitation in the opposite one. The equatorial injections in turn result in an overall reduction of tropical precipitation. Changes at mid and high latitudes, which are much smaller in absolute terms than those in the tropics, can be as high in terms of percentage changes as those in the tropics, but the models disagree on the sign and magnitude of the responses there far more than they do in the tropics. We also show % in order to highlight the significance of the absolute changes based on the overall received precipitation. This for instance helps to show that, compared to the baseline values, the very small changes observed at high latitudes may also be significant.

In general, precipitation changes depend on even more factors than temperature changes (Kravitz et al., 2013; Tilmes et al., 2013). Differences in the inter-hemispheric cooling patterns produce shifts in the precipitation by moving the intertropical convergence zone (ITCZ, see below and e.g. Ridley et al. (2015)); this is compounded with differences in the cloud responses (Smyth et al., 2017) and by the large-scale dynamical changes resulting from the aerosol-induced stratospheric heating (see Sections 3.2 to 3.5 in PART2; also Simpson et al. (2019)). Furthermore, precipitation responses tend to be regionally dependent: for instance, previous G6 GeoMIP simulations showed consensus in the winter precipitation response over Europe (with increased precipitation in northern Europe and decreased over southern Europe associated with the SAI-induced positive phase of the North Atlantic Oscillation); however, the same models showed little consensus over the North American continent (Jones et al., 2022).

Figure 9 shows changes in the position of ITCZ, which we define here as the latitude near the equator where the meridional mass streamfunction at 500 hPa changes sign. The models qualitatively agree with respect to the simulated ITCZ shifts, despite the significant differences in the climatological ITCZ locations (shown on the left size of Fig. 9). The larger simulated ITCZ shifts for the NH injections compared to the SH injections are also consistent with the different cooling patterns, with stronger and more NH-confined cooling for the 30°N and 15°N injections. A similar behavior of the ITCZ was also previously found for volcanic eruptions occurring in the equatorial or extra-equatorial regions (Trenberth and Dai, 2007; Oman et al., 2006; Haywood et al., 2013; Ridley et al., 2015) as well as for previous single-point SAI simulations (Haywood et al., 2013).

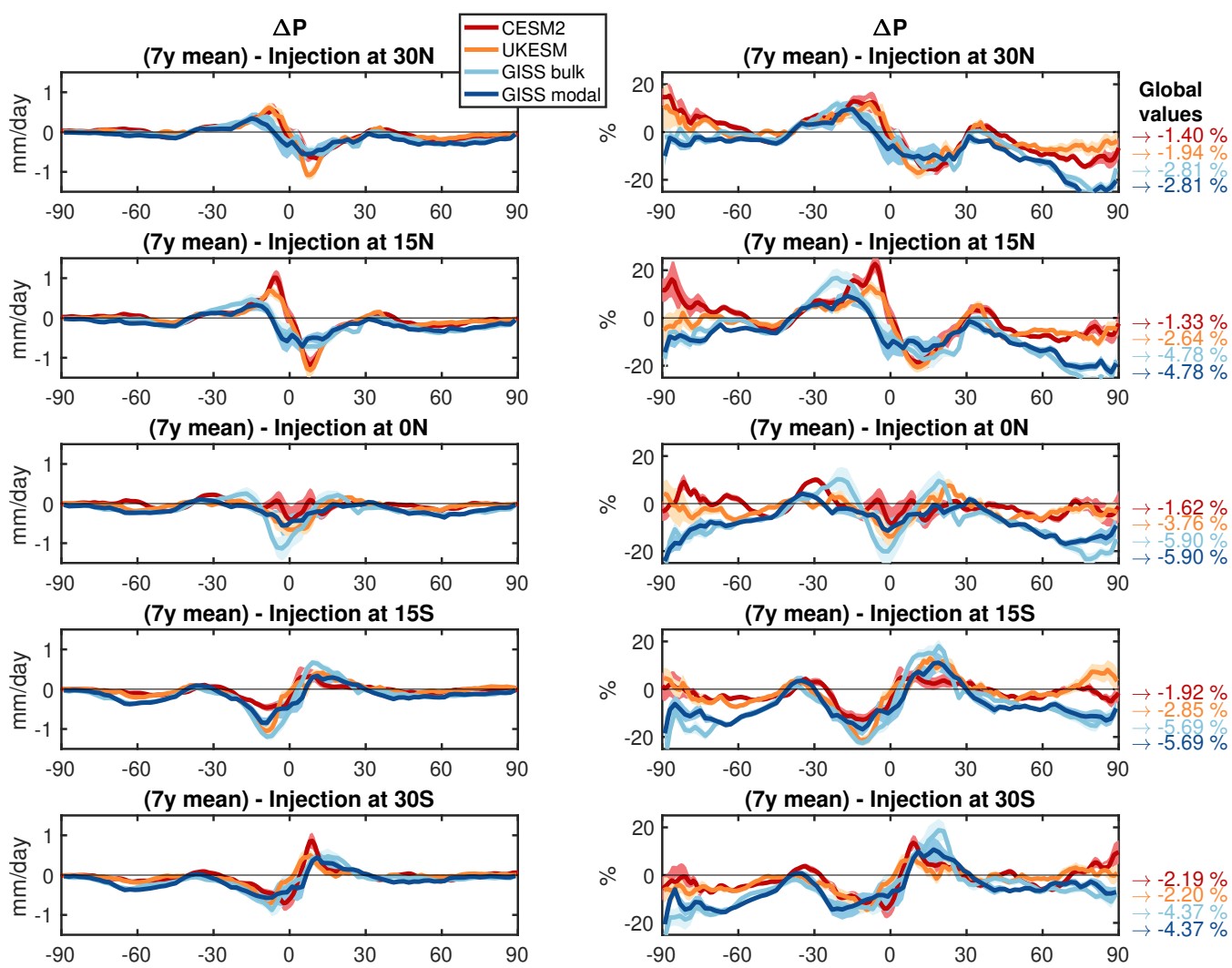

**Figure 8.** (a-e) Zonal and annual mean changes in precipitation (mm/day) resulting from the single-point injections at either 30°N (a), 15°N (b), 0°N (c), 15°S (d) or 30°S (e), averaged over the last seven years of simulation. (f-j) As (a-e), but with percent changes obtained by normalizing to the baseline zonal mean precipitation in each model.

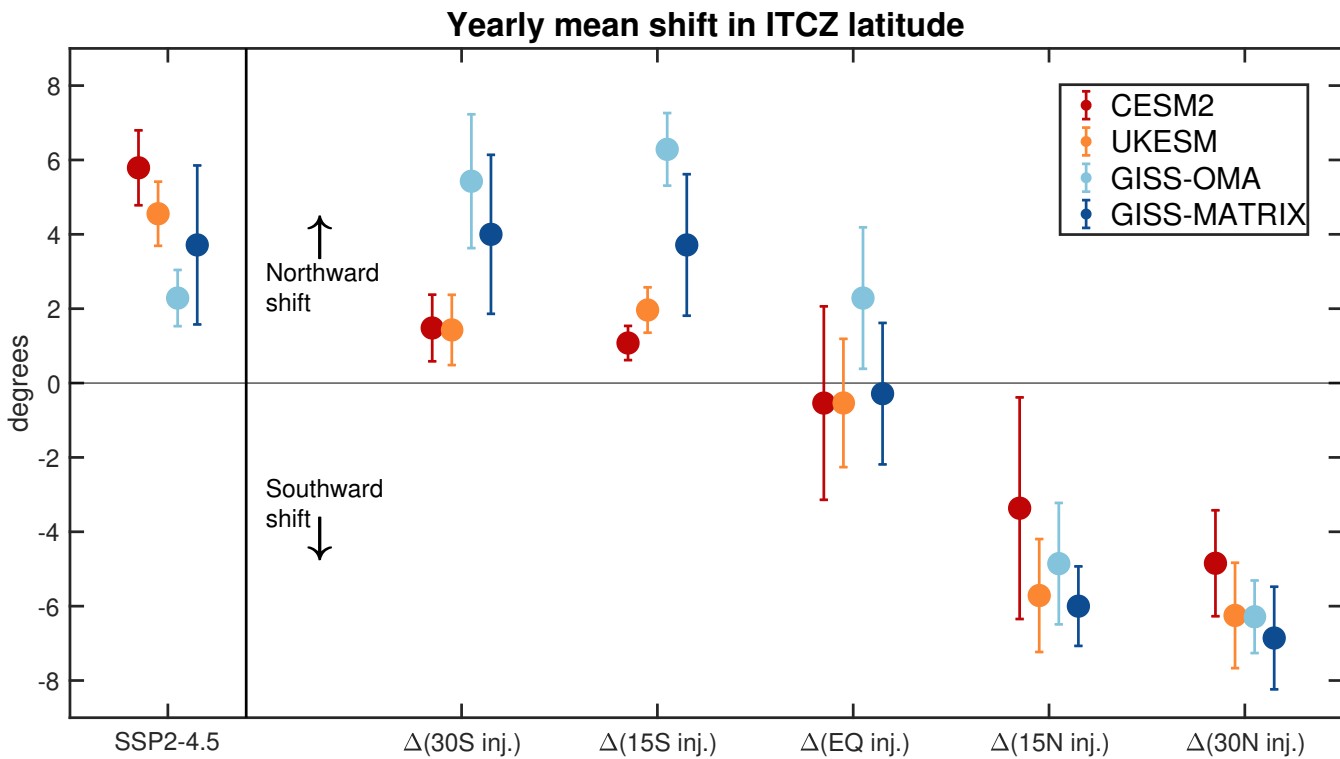

**Figure 9.** Annual mean changes in the latitude of ITCZ, here defined as the latitude near the equator where the meridional mass stream-function at 500 hPa changes sign (defined as the mid-point between the two gridboxes). Results are shown for each injection location (30°S, 15°S, 0°N, 5°N and 30°N) and each model. Error bars indicate the $\sigma$ standard deviation for the last seven years of the SAI simulation. Shown also on the left hand side is the climatological ITCZ position in the corresponding SSP2-4.5 simulation.

## 5 From single point injections to multi-target simulations: designing the feedback algorithm

One of the goals of this exercises is to reproduce in multiple climate models a similar geoengineering strategy to that described in Kravitz et al. (2017), i.e. where a feedback algorithm determines the amount of $SO_2$ to be injected each year at each of the specified locations in order to achieve a set of predetermined surface temperature targets. As discussed in Kravitz et al. (2016) and Zhang et al. (2022), year-round injections at the four injection locations described here ($30°N$, $15°N$, $15°S$ and $30°S$) may allow for the management of three independent degrees of freedom in surface temperature (global mean temperature, temperature difference between the two hemisphere, temperature difference between the pole and the tropics). Combining varying rates of injections at these four locations produces distinct patterns of AOD.

One method of describing and decomposing these AOD patterns into simpler ones is to project the zonal mean AOD onto the first three Legendre polynomials evaluated as a function of sine of latitude (hereafter, $\psi$), which equate to three idealized patterns of optical depth achievable. Here $\ell_0 = 1$ indicates a pattern of AOD uniform at all latitudes, which therefore produces a uniform cooling; $\ell_1 = sin(\psi)$ indicates a pattern of AOD that is greater in one hemisphere and smaller in another, which therefore can cool one hemisphere more than another; and $\ell_2 = \frac{1}{2}(3sin^2(\psi) - 1)$ indicates a quadratic pattern that is smaller close to the equator and greater at high latitude, therefore cooling those latitudes more. MacMartin et al. (2017) demonstrated that these three patterns are achievable through the combination of injections at different locations: injecting at the same time at $15°N$ and $15°S$ produces $\ell_0$; injecting at $15°$ and $30°$ N (or S) produces $\ell_0 + \ell_1$ ($\ell_1^N$, more AOD in the northern hemisphere) or $\ell_0 - \ell_1$ ($\ell_1^S$, more AOD in the southern hemisphere), respectively; and injecting at $30°N$ and $30°S$ produces $\ell_0 + \ell_2$. Therefore, injections at these four latitudes can be combined (assuming linearity) to achieve desired combinations of $\ell_0$, $\ell_1$, and $\ell_2$, subject to the constraints $\ell_0 \geq 0$, $\ell_2 \geq 0$, and $\ell_0 \geq |\ell_1| + \ell_2$. These constraints derive from the fact that a negative injection (and a negative pattern of AOD) are not possible.

This truncated Legendre decomposition of the zonal mean AOD into $\ell_0$, $\ell_1$, and $\ell_2$ is not the only way to numerically define an SAI strategy, but it has been found to be sufficient to independently control the three degrees of freedom in surface temperature: the global mean temperature, $T_0 = \frac{1}{A} \int_\psi T(\psi) \cos(\psi) d\psi$ (where $T(\psi)$ is the zonal mean surface temperature and $A = \int_\psi \cos(\psi) d\psi$ is a normalization factor), controlled by $\ell_0$; the pole-to-pole temperature gradient $T_1 = \frac{1}{A} \int_\psi \sin(\psi) T(\psi) \cos(\psi) d\psi$, controlled by $\ell_1$; and the equator-to-pole temperature gradient $T_2 = \frac{1}{A} \int_\psi \frac{3\sin^2(\psi)-1}{2} T(\psi) \cos(\psi) d\psi$, controlled by $\ell_2$. Similarly, the definition of the four injection latitudes described here is not exhaustive, in the sense that it is not the only way to produce the three described aerosol patterns, as there could be other latitudes of injection that once combined could produce $\ell_0$, $\ell_1$ and $\ell_2$ with smaller residuals. Our choice is thus only intended to illustrate the optimization method.

In this section, we present the calculations necessary for this strategy to be implemented in the three climate models considered in this study; similar analyses for CESM1(WACCM) are performed and explained in detail in MacMartin et al. (2017). In order to design such a feedback control algorithm, it is first necessary to quantify the relationships between injections at each

of the four latitudes ($q = [q_{30S}\ q_{15S}\ q_{15N}\ q_{30N}]^T$, where $q$ is the amount of injection per year in Tg-SO$_2$) and the patterns of AOD they create ($\ell = [\ell_0\ \ell_1^N\ \ell_1^S\ \ell_2]^T$), and then the relationship between AOD ($\ell$) and temperature ($T_0$, $T_1$, and $T_2$). The first relationship can be defined by two matrices M and F such that $q = MF^{-1}\ell$. For each the four models, using the values shown in Figure 2, this relationship is computed to be as follows:

$$
CESM: \begin{bmatrix} q_{30S} \\ q_{15S} \\ q_{15N} \\ q_{30N} \end{bmatrix} = \begin{bmatrix} 25\ell_1^S + 38\ell_2 \\ 39(\ell_0 - (\ell_1^N + \ell_1^S + \ell_2)) + 45\ell_1^S \\ 34(\ell_0 - (\ell_1^N + \ell_1^S + \ell_2)) + 59\ell_1^N \\ 15\ell_1^N + 43\ell_2 \end{bmatrix}
\quad
UKESM: \begin{bmatrix} q_{30S} \\ q_{15S} \\ q_{15N} \\ q_{30N} \end{bmatrix} = \begin{bmatrix} 56\ell_1^S + 50\ell_2 \\ 37(\ell_0 - (\ell_1^N + \ell_1^S + \ell_2)) + 22\ell_1^S \\ 36(\ell_0 - (\ell_1^N + \ell_1^S + \ell_2)) + 18\ell_1^N \\ 49\ell_1^N + 46\ell_2 \end{bmatrix}
$$

$$
GISS_{OMA}: \begin{bmatrix} q_{30S} \\ q_{15S} \\ q_{15N} \\ q_{30N} \end{bmatrix} = \begin{bmatrix} 6\ell_1^S + 26\ell_2 \\ 23(\ell_0 - (\ell_1^N + \ell_1^S + \ell_2)) + 35\ell_1^S \\ 20(\ell_0 - (\ell_1^N + \ell_1^S + \ell_2)) + 44\ell_1^N \\ 0\ell_1^N + 29\ell_2 \end{bmatrix}
\quad
GISS_{MATRIX}: \begin{bmatrix} q_{30S} \\ q_{15S} \\ q_{15N} \\ q_{30N} \end{bmatrix} = \begin{bmatrix} 38\ell_1^S + 46\ell_2 \\ 44(\ell_0 - (\ell_1^N + \ell_1^S + \ell_2)) + 47\ell_1^S \\ 39(\ell_0 - (\ell_1^N + \ell_1^S + \ell_2)) + 69\ell_1^N \\ 22\ell_1^N + 49\ell_2 \end{bmatrix}
$$

$$(4)$$

These values can be obtained considering separately the injection amounts necessary to obtain the four AOD patterns described above, and solving the linear least-squares problem considering as a constraint that no negative injection, or AOD values, are possible. This is shown in Figure 10 for all models, with the dashed black lines being the idealized patterns; the linear combination of the coupled injection locations in each of the four cases, which results in the colored lines in each plot, successfully manages to produce a pattern similar to the idealized one, indicating that by injecting in the locations presented in the tables below each panel the quantities indicated, those patterns are achievable. Models all agree on the need to deploy quasi-symmetrical amounts to obtain the $\ell_0$ and $\ell_0 + \ell_2$ patterns (with at most a 20% difference in some cases) whereas they differ largely on how to obtain the $\ell_0 + \ell_1$ and $\ell_0 - \ell_1$ patterns. For instance, to obtain $\ell_0 + \ell_1$, CESM2 requires four times as much injection at 15°N than it does at 30°N, whereas UKESM1 requires two times as much injection at 30°N than it does at 15°N, and GISS requires no injection at 30°N at all. The residuals (the difference between the solution of the linearized system and the idealized pattern) also show that how close the models get to the idealized patterns differs, with CESM2 being twice as low as any other models in all cases. It is important to note that the results obtained with the current version of CESM2 (CESM2-WACCM6) are very similar but not identical ($< 10\%$) to those obtained and discussed in MacMartin et al. (2017) using CESM1(WACCM). In that case, injections were at different altitudes (25 km for 15°N and S, and 22 km for 30°N and S), and so some differences are to be expected.

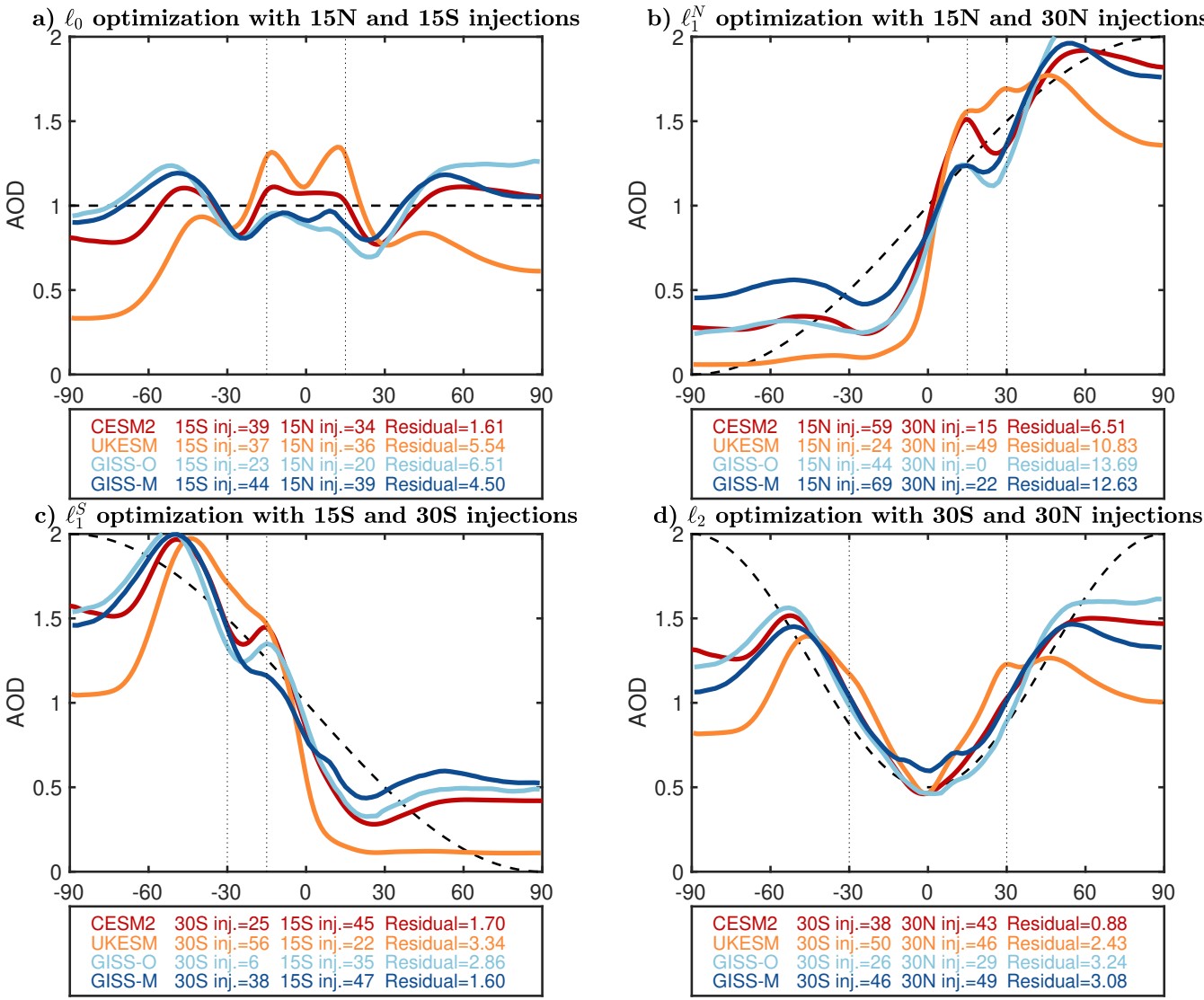

**Figure 10.** Linear least-squares solution to obtain an $\ell_0$-shaped AOD (black dashed line) using only $15°$N and $15°$S injections (a), an $\ell_0 + \ell_1$ ($\ell_1^N$, black dashed line, increased AOD in the Northern Hemisphere) one using only $15°$N and $30°$N injections (b), an $\ell_0 - \ell_1$ ($\ell_1^S$, black dashed line, increased AOD in the Southern Hemisphere) one using only $15°$S and $30°$S injections (c) and a $\ell_2$ (black dashed line, increased AOD at high latitudes compared to low latitudes) one using only $30°$S and $30°$S injections (d). Values at the bottom of each plot (in Tg-$SO_2$/yr) indicate the required injection, assuming linearity, based on the AOD distribution in Figure 2. The residual of each solution is also included.

Using the patterns of temperature response shown in Figure 6, combined with the patterns in AOD, we can derive the second set of relationships, between the AOD projections $\ell$ and $T_0$, $T_1$ and $T_2$. For the four models, this leads to

$$
CESM : \begin{bmatrix} T_0 \\ 3T_1 \\ 5T_2 \end{bmatrix} \simeq \begin{bmatrix} -\mathbf{4.1} & 0 & 0 \\ -3.0 & -\mathbf{3.9} & 0 \\ -1.5 & -1.6 & -\mathbf{0.5} \end{bmatrix} \begin{bmatrix} \ell_0 \\ \ell_1 \\ \ell_2 \end{bmatrix} \quad UKESM : \begin{bmatrix} T_0 \\ 3T_1 \\ 5T_2 \end{bmatrix} \simeq \begin{bmatrix} -\mathbf{4.6} & 0 & 0 \\ -2.6 & -\mathbf{3.6} & 0 \\ -0.2 & -1.5 & -\mathbf{1.3} \end{bmatrix} \begin{bmatrix} \ell_0 \\ \ell_1 \\ \ell_2 \end{bmatrix}
$$

$$
GISS_{OMA} : \begin{bmatrix} T_0 \\ 3T_1 \\ 5T_2 \end{bmatrix} \simeq \begin{bmatrix} -\mathbf{5.6} & 0 & 0 \\ -0.9 & -\mathbf{4.4} & 0 \\ 0.7 & -1.0 & -\mathbf{4} \end{bmatrix} \begin{bmatrix} \ell_0 \\ \ell_1 \\ \ell_2 \end{bmatrix} \quad GISS_{MATRIX} : \begin{bmatrix} T_0 \\ 3T_1 \\ 5T_2 \end{bmatrix} \simeq \begin{bmatrix} -\mathbf{9.2} & 0 & 0 \\ -4.2 & -\mathbf{6.8} & 0 \\ -1.4 & -2.4 & -\mathbf{6} \end{bmatrix} \begin{bmatrix} \ell_0 \\ \ell_1 \\ \ell_2 \end{bmatrix} \quad (5)
$$

This result indicates that the main perturbations produced on $T_0$, $T_1$, and $T_2$ result from the respective AOD patterns, as the larger numbers are on the main diagonal. A controller algorithm based on the values shown in equations 4 and 5 can, in a strategy that aims to achieve a certain climatic target, read the annually averaged surface temperatures at the end of one year,

determine the distance between those and the target ones and determine how the cooling can be achieved through the three AOD patterns following Equation 5. Then, with Equation 4, the AOD pattern necessary can be related to four different amount of injections at the four locations. For CESM2, these results have already been applied to produce two datasets (described in Richter et al. (2022) and MacMartin et al. (2022)), and similar efforts are ongoing for UKESM and GISS.

Overall, the results shown in this section confirm that the methods applied in MacMartin et al. (2017) to produce the results in Kravitz et al. (2017) may be applicable to other climate models as well, and that the inter-model differences in transport and microphysics can, in this kind of optimization strategy, be overcome by modifying the ratio between injection amounts at different locations. However, it is also clear that, as indicated by the differences in the residuals shown in Fig. 10, the use of the specific four latitudes proposed in MacMartin et al. (2017) might not result to be the most optimal set of latitudes in all

models, depending on the specifics of the stratospheric circulation. The results of UKESM for instance suggest that, in that model, the 15°N/15°S are too close to the tropical pipe and therefore still confine the aerosols too much at low latitudes, as opposed to CESM2 and GISS. Therefore, based on UKESM results, a choice of latitudes further from the equator (such as, for instance, 20°) may result in a better optimization. Similarly, all models seem to suggest that the successful obtainment of $\ell_2$ might be hard to achieve with only 30° injections, and that some higher latitude injections might be necessary. It is thus worth

reiterating that the specific choices of latitudes here should not be read as the only, nor optimal, choice, but merely as a way to start exploring the available space (see also Zhang et al. (2022) for a broader exploration of this issue).

## 6    Conclusions

In this work, we have shown the results of a first systematic intercomparison of climate responses to fixed single point $SO_2$ injections in different climate models with interactive aerosol microphysics and comprehensive stratospheric chemistry. In

particular, we used CESM2-WACCM6, UKESM1.0, GISS-E2.1-G with bulk aerosol microphysics and GISS-E2.1-G with

two-moment aerosol microphysics, and a set of simulations injecting a fixed quantity of $SO_2$ at five different latitudes (30°N, 15°N, 0°N, 15°S and 30°S) and at 22 km of altitude. The same protocol was used in all models to determine similarities and differences in the resulting stratospheric sulfate aerosol distributions, alongside in the resulting atmospheric and surface climate responses. Similar simulations had previously only been performed with one climate model, CESM1(WACCM) (MacMartin et al., 2017). Our multi-model simulations therefore serve a dual purpose: i) to evaluate the responses to off-equatorial $SO_2$ injections in multiple climate models, understanding similarities and differences between different independent climate models, and, thus, to isolate sources of uncertainty in model SAI responses and identify future areas of improvements; and ii) to lay the basis for a future intercomparison between models using a feedback algorithm capable of achieving multiple temperature targets (MacMartin et al., 2017) similar to the one used in Tilmes et al. (2018a) to produce the Geoengineering Large Ensemble.

The simulated changes in stratospheric Aerosol Optical Depth were analyzed in terms of both their absolute magnitudes and the values normalized per unit of global mean AOD achieved. The latter assumes the the latitudinal distribution of simulated AOD remains similar under varying injection rates. Previous analyses of the GLENS results showed that this assumption holds, unless very large injections are considered that might noticeably modify the stratospheric circulation and prevent the aerosols from reaching the higher latitudes (Visioni et al., 2020). Our results in this work showed that in order to achieve similar global AOD values, the models considered would require different amount of Tg-$SO_2$/yr injected, with the largest differences for the model with a different, simpler, aerosol treatment (GISS-E2.1-G with bulk microphysics). By analyzing the results in terms of the normalized distributions, which bypasses the bias due to different simulated global mean AOD values under the same injection rate, the largest inter-model differences were found in the tropics (especially for the equatorial injections) and at very high latitudes, with models disagreeing on the amount of aerosols transported poleward.

Using a similar separation between the absolute magnitudes and the normalized latitudinal changes we analyzed the resulting surface temperature responses. As before, a large discrepancy between the two GISS versions and the other two models was identified for the global mean response, with the former showing a global cooling per unit AOD roughly a factor of two (for the bulk aerosol version) or three (for the modal aerosol version) times larger than that simulated by either CESM2-WACCM6 and UKESM1.0. When the surface temperature responses are normalized, however, models generally show a good agreement amongst them in terms of the overall latitudinal distribution of the temperature changes, and similarly they show a good agreement in the response of the latitudinal changes in precipitation.

Uncertainties in the projected SAI responses closely track those reported for simulations of past explosive volcanic eruptions (Clyne et al., 2021). Our results show that a large fraction of these uncertainties arise from discrepancies in the $SO_2$-to-sulfate aerosol conversion and their subsequent growth in the three models with three different model aerosol schemes, although differences in the stratospheric circulation also play an important role, as shown in PART2. In addition, comparison of the two sets of GISS simulations using either two-moments or bulk treatment of aerosol microphysics showed important differences in terms of the simulated AOD and the overall cooling produced per unit AOD (Figure 7), as well as in terms of the resulting

stratospheric response (PART2), highlighting the importance of detailed treatment of microphysical processes. In agreement, a recent study from Laakso et al. (2022) using ECHAM-HAMMOZ showed that the choice of aerosol scheme (modal versus sectional) can lead to large discrepancies in the resulting radiative forcing due to the injection of different quantities of $SO_2$ (up to twice as large for some scenarios). Our GISS results here further confirm that the choice of aerosol microphysical scheme, all else being equal, can influence the overall amount of cooling, although the results are still even more influenced by the choice of climate model.

This work has shown that it would be feasible to replicate a global-scale injection strategy, such as the one used in GLENS, in multiple climate models; if the amount of $SO_2$ injected can be controlled year by year, models appear capable of reproducing a similar scaled surface temperature response. This thus ensures that a strategy including injections at 30°N, 15°N, 15°S and 30°S would potentially be able to maintain the three defined temperature targets (i.e. global mean surface temperature, and equator-to-pole and inter-hemispheric surface temperature gradients), albeit with differences in the injection magnitudes between models. A future comparison of the results of such an experiment using different climate models maintaining similar temperature targets would help identify different sources of uncertainty in the modelled response to SAI as compared to fixed-point injection simulations, for instance regarding the behavior of the Atlantic Meridional Overturning Circulation (Tilmes et al., 2020), the North Atlantic Oscillation (Jones et al., 2022) or impacts on middle atmospheric composition, including stratospheric ozone (Tilmes et al., 2021). When it comes to climate impacts, focusing on particular large-scale temperature targets rather than analysing the direct outcomes of fixed single-point injections shifts, in a way, the focus from one kind of uncertainty to another. If the location and amount of injections can be modified, some of the stratospheric uncertainties discussed here would matter less, as the strategy could be adjusted to obtain a pattern of aerosol distribution functional to obtaining a certain pattern of cooling. The consistency in the normalized temperature response between models indeed seems to suggest this is possible. If similar large-scale patterns of cooling can be achieved consistently between models, more focus can then be given to understanding the uncertainties in the projected regional scale responses, which are an essential to properly assess local risks and adaptation strategies. This does not remove uncertainties related to the large scale transport of the aerosols, or those arising from any indirect drivers of the surface responses, like the associated changes in concentrations of lower stratospheric ozone and water vapour alongside any changes in circulation; but it suggests that, with this strategy, they can be more easily separated, as we illustrate more in depth in PART2 using these same simulations. Due to the potential for our proposed experiments to shed further light on processes relevant for SAI, and as a fundamental experiment to develop an intercomparison of more comprehensive SAI strategies, we would like to suggest them also as a possible Testbed experiment for the Geoengineering Model Intercomparison Project.

*Code availability.* The code used to calculate the matrices and figures in Section 5 is available at https://github.com/dan-visioni/code-for-gains-calculator.

*Data availability.* The data used in this work will be available in the Cornell eCommons repository upon publication of the preprint, and the link will be added upon publication. Please contact DV to access the data before that.

*Author contributions.* DV performed the CESM2 simulations, wrote the manuscript and performed the analyses. EB performed the analyses
and contributed to the manuscript. WRL performed the analyses and wrote the public code for Section 5 with help from DGM. AJ and JH performed the UKESM simulations. BK performed the GISS simulations. DGM contributed to the manuscript.

*Competing interests.* The authors declare no competing interests

*Acknowledgements.* Support was provided by the Atkinson Center for a Sustainable Future at Cornell University for DV, EMB and DGM. Support for B.K. was provided in part by the National Science Foundation through agreement CBET-1931641, the Indiana University En-
vironmental Resilience Institute, and the *Prepared for Environmental Change* Grand Challenge initiative. The Pacific Northwest National Laboratory is operated for the US Department of Energy by Battelle Memorial Institute under contract DE-AC05-76RL01830. AJ and JMH were supported by the Met Office Hadley Centre Climate Programme funded by BEIS and by SilverLining through its Safe Climate Research Initiative.

The Community Earth System Model (CESM) project is supported primarily by the National Science Foundation. We would like to acknowledge high-performance computing support from Cheyenne (https://doi.org/10.5065/ D6RX99HX) provided by NCAR's Computational and Information Systems Laboratory, sponsored by the National Science Foundation. The UKESM simulations were carried out using MONSooN2, a collaborative High-Performance Computing facility funded by the Met Office and the Natural Environment Research Council. NASA GISS ModelE simulations were supported by the NASA High-End Computing (HEC) Program through the NASA Center
for Climate Simulation (NCCS) at Goddard Space Flight Center. Data storage at Indiana University is supported by the National Science Foundation under Grant No. CNS-0521433.

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
