# Peer review of "Climate response to off-equatorial stratospheric sulfur injections in three Earth System Models - Part 1: experimental protocols and surface changes"

_EGUsphere, 2022_

## Author Comment (AC1)

Response to reviewer 1

**Reviewer comments are in bold**, authors' responses are in blue. Yellow: to do

**In this study, the authors investigate how stratospheric aerosol intervention (SAI) using SO2 injections at different latitudes affects the aerosol distribution, aerosol optical depth, and surface climate (temperature and precipitation) in three different Earth System Models. The authors find differences between the models and also between different aerosol setups in the same model. The authors then describe the development of feedback algorithms to be used in future simulations to manage injections of SO2 to meet temperature targets as the runs proceed. In general, the paper is clear and easy to read, and the analysis is logical. The results will be of interest to the geoengineering community and the paper is well-suited to ACP. I have two main comments that I suggest the authors consider before publication:**

We thank the reviewer for their helpful and supportive comments. We have addressed all points below and modified the manuscript accordingly.

**Given the focus on aerosol microphysics driving differences between these results, can the authors highlight how the aerosol schemes differ between the models, not just in terms of the modal properties (Table 1), but how the aerosol processes are treated? It's mentioned on L145 that condensation is treated differently in GISS, but how? Given the differences found for effective radius, it would be useful to show (perhaps in the SI) some of the other aerosol metrics such as SO2 conversion (highlighted on L419 as an important discrepancy), the nucleation, condensation and coagulation rates, and fluxes between the modes, and explain how these parameterizations differ between the models. Can we learn more here about these uncertainties compared to multi-model volcanic eruption studies that have already shown that differences in AOD are due to different microphysical schemes? What specific areas of improvement have been found in this work as stated in the conclusions at L395?**

Thank you for the insightful suggestion. We have updated the manuscript expanding on models' differences in terms of their microphysical schemes. The models unfortunately do not output many of the requested variables, so it is difficult to ascertain exactly which process is the dominant contributor to uncertainties. Nevertheless, all of these schemes have been validated to some degree against volcanic eruptions (see references in the text).

**Section 5 - without going back to previous references (such as Kravitz et al., 2017), this section was hard to follow, especially for someone not familiar with such feedback algorithms. It would be useful if the mathematical relationships were described further in the text and that all letters and symbols were defined and listed immediately after the equations – for example, q and equations for T0 – T2. It would also be helpful if the section was more explicit with signposting to the relevant subplot or line on Figure 10 – e.g., expanding L374 to 'pattern 'of AOD' similar to the target (black dashed lines)', or**

**similar. It was also not clear to me how the feedback algorithms are different/similar to previous work and what the implications are.**

Based on this comment from both reviewers, we have modified Section 5 to further clarify all aspects of this portion of the work.

**Specific comments**

**Abstract: an extra sentence at the end summarizing the overall implications of this work would be useful. It was also not clear whether all models included modal aerosol microphysics schemes. I would suggest introducing the 4 model setups at the start**

We have added a phrase at the very beginning of the abstract stating:

"Here we present the results from the first systematic intercomparison of climate responses in three Earth System Models where the injection of SO2 occurs at different latitudes in the lower stratosphere: CESM2-WACCM6, UKESM1.0 and GISS-E2.1-G. The first two, and a version of the third, use a modal aerosol microphysics scheme, while a second version of GISS-E2.1-G uses a bulk aerosol microphysics approach."

And a phrase at the end:

"In conclusion, we demonstrate that it is possible to use these simulations to produce more comprehensive injection strategies, large differences in the injection magnitudes can be expected, potentially increasing inter-model differences in the stratosphere while reducing surface ones."

**L30-L35: A few more relevant references could be added here e.g., Zanchettin et al. (2016; 2022).**

Thank you for the suggestion! We have added these references.

**L82: It was unclear at this stage what these targets are**

We have specified these are the same targets we referred to in the previous paragraph.

**L121: What are the differences in the aerosol scheme?**

We have tried to discuss more in depth some of the differences between the schemes.

**L149: Please clarify what you mean here**

We meant that the value reported in the paper has been subsequently updated (but no publications have reported it). We have added a "used in the current version of GISS used here" to the phrase.

**L181: Why AOD and not SAOD? Please also describe the overall evolution of this figure – i.e., the ~2 years of adjustment and therefore why the last 7 years are used in subsequent averages**

We have fixed the figure by modifying the title and adding a dashed line to indicate the considered steady state and explained it in the caption.

**L235: How is the lifetime defined? Please remove 'obviously'**

We have modified the phrase as follow, for clarity:

"Finally, panels f-h in Figure 4 give an overview for all cases of the ratio between the overall mass of the produced aerosols (shown alone in panel 4f) and the injected amount of SO2. This ratio, shown in panel 4g, represents the lifetime of the added sulfate: as we are in a steady state, where no new mass is added to the global burden, this lifetime can be calculated as the burden divided by its constant source (the injection) (Visioni et al., 2018)"

**L253: Is this wet or dry radius? Fig. SX --> Fig. S1.**

That's wet radius. Thanks for spotting the mistake!

**L263: What are the dynamical differences? Do the authors have an explanation for the stronger poleward transport in CESM given also the differences in particle size? Why is the transport in the 15S case more similar to GISS modal?**

The dynamical differences are discussed in depth in the companion paper. We have added further clarification in this phrase:

"... these differences are discussed more in depth in PART2 and indicate a substantially stronger climatological shallow branch of the Brewer-Dobson circulation in CESM2 compared to other models (see Figure 2 in PART2)"

**L279: The initial results shown between the global mean AOD and global mean temperature in Figure 1 could also be discussed here.**

Thank you for the suggestion, we have added a comment about Fig.1 here.

**L289 – L305: I found this hard to follow. Has the sensitivity to aerosols in GISS been increased or not?**

The reviewer is right and the phrase was incredibly confusing. Apologies. We have rephrased below:

*"As noted for Fig. 1, this large normalized difference is mainly due to differences in the global cooling produced between the two GISS realizations (the sensitivity to the aerosols amount); even if there is a much smaller amount of AOD simulated in GISS modal, it results in a similar level of surface cooling as is simulated in GISS bulk under much larger AOD values."*

**L310-311: Unclear exactly what you mean here**

We have modified the phrase:

*"In contrast, the models tend to disagree with respect to responses in the NH high latitudes much more than for the SH high latitudes; this can be explained by the role of dynamical atmospheric variability and sea-ice variability, which in the NH both contribute to the overall response increasing model disagreement"*

**Figure 7: What's causing the different response for CESM2 and UKESM for 30N compared to the other injection locations?**

The differences in Fig. 7 for 30N appear to be in line with those at other injection locations, so we're not sure what the reviewer means here.

**L314: I would suggest moving the overall description of the precipitation changes from the second paragraph to here as it is a long time before the results are described. What about the global percentage changes shown in Figure 8?**

Thank you for the suggestion, we have done so. We have also added a phrase for the % changes:

*"We also show \% in order to highlight the significance of the absolute changes based on the overall received precipitation. This for instance helps to show that, compared to the baseline values, the very small changes observed at high latitudes may also be significant."*

**L336: shown on left hand side of Figure 8?**

Thank you for the suggestion, added.

**L338: There are several newer studies on the impact of eruptions on the ITCZ that could be cited here. Please see Marshall et al. (2022) for some examples.**

Thank you for suggesting this, we have added some more recent references here.

**Figure 10 caption: Please explain what L0, L1 and L2 are and label the black dashed lines.**

Done.

**L346: I think it would be helpful to state what these are here, as is done in the conclusions**

Agreed!

**L399: This paragraph focuses on the methods, but what are the actual results? How do the results differ depending on the injection location?**

We have included a final paragraph discussing this issue: We report it here:

"*Overall, the results shown in this section confirm that the methods applied in MacMartin et al. (2017) to produce the results in Kravitz et al. (2017) may be applicable to other climate models as well, and that the inter-model differences in transport
and microphysics can, in this kind of optimization strategy, be overcome by modifying the ratio between injection amounts at different locations. However, it is also clear that, as indicated by the differences in the residuals shown in Fig. 10, the use of
the specific four latitudes proposed in MacMartin et al. (2017) might not result to be the most optimal set of latitudes in all models, depending on the specifics of the stratospheric circulation. The results of UKESM for instance suggest that, in that
model, the 15°N/15°S are too close to the tropical pipe and therefore still confine the aerosols too much at low latitudes, as opposed to CESM2 and GISS. Therefore, based on UKESM results, a choice of latitudes further from the equator (such as, for instance, 20°) may result in a better optimization. Similarly, all models seem to suggest that the successful obtainment of L2 might be hard to achieve with only 30° injections, and that some higher latitude injections might be necessary. It is thus worth reiterating that the specific choices of latitudes here should not be read as the only, nor optimal, choice, but merely as a way to start exploring the available space (see also Zhang et al. (2022) for a broader exploration of this issue)*"

**L447: This last sentence is difficult to follow**

Thanks, we have changed it.

**Technical corrections**
We have fixed all of these, thank you!

**L3: occurs**

**L23: please add numbers for the three items in this list**
**L69: only --> one**
**L71: a --> the**
**L84: impacts**
**L86: in --> with?**
**L120: eruptions**
**L208: shows**
**L209: standard deviations**
**L211: check commas**
**Figures: please check all x and y labels are present (missing from 2, 6 and 8) and remove red/green line combinations**

We have changed the colour schemes and improved all the figures also based on other feedbacks.

**L312: 2020 --> 2021**
**L318: clouds --> cloud**
**L320: is --> are**
**Figure 8 caption: five --> seven**
**L441: insert 'than'**
**L444: seems**

---

## Author Comment (AC2)

Response to reviewer 2

**Reviewer comments are in bold**, authors' responses are in blue.

**The authors compare the output of geoengineering simulations performed with three Earth's system models (1 ran with two different aerosol schemes) to determine the difference in AOD, temperature, and precipitation response produced using the same injection of SO2. The authors provide an exhaustive comparison of these quantities (especially AOD and temperature, less so precipitation) and attempt to provide a hypothesis about the reasons for discrepancies.**

We thank the reviewer for their comments. We respond to the points raised below..

**Generally, I have found this article clear, with a good choice of figures, but the complete lack of observations limits its impact. Of course, I am aware that there are no observations of geoengineering, but variables to evaluate, for instance, the isolation of the tropical stratosphere or the background (non-SAI) AOD can be evaluated against observations. Introducing observations would allow understanding which model has a more reliable representation of transport and dynamics, as well as of background aerosol and sensitivity to changes. I understand that this evaluation against observations is not the focus of this paper, but it has been probably (hopefully?) done in other articles and the main findings could be reported here. Otherwise, the main message of this paper is "the models differ", which is for sure correct but not particularly telling unless we can understand whether all of these models produce equally possible outcomes or if one is less reliable than the others.**

We thank the reviewer for their comments. An evaluation of baseline circulation is the subject of Part 2 of this study, where its role in contributing to the simulated aerosol distributions is also discussed in depth. We have taken care to reference the relevant parts of PART2 frequently throughout the discussions of the results in Part 1 (as also suggested by reviewer 1); we have also added some other references of past evaluations of stratospheric circulation.
For the AOD evaluation, we have also added some more references: however, the background stratospheric OD doesnt really offer a reliable estimate of microphysical growth under conditions with far more sulfates. We have included in the revised manuscript some further comments on the availability and future necessity of comparisons with previous volcanic eruptions.

**Secondarily, I am not sure if the OMA experiment has been set up correctly. I couldn't find anywhere how the aerosol radius was chosen. Is it the usual radius used for tropospheric aerosol? It seems like most of the differences between OMA and MATRIX result from a much smaller aerosol radius than the other models. The authors should have first run an experiment with MATRIX, calculated the resulting effective radius, and set up OMA to have that effective radius. As it is I am not sure about the significance of the OMA experiment.**

It is the usual radius used for tropospheric aerosol, as there is only one place to specify the radius. We have experience with simulations in which we increased this radius to better match the stratospheric aerosol size (Pitari et al., 2014), and the tropospheric aerosol size becomes unrealistically large, which has non-negligible effects on radiative forcing, tropospheric chemistry, and deposition. While we like the approach suggested by the reviewer, it too has tradeoffs, and there isn't really a best way to proceed with the OMA configuration. We have included it anyway as an interesting comparison, but we agree that its usefulness is limited. We have now articulated this better in the manuscript. At section 2.4 in particular, we have added what follows:

"*We note that in the GISS with bulk treatment, there is one specified aerosol dry radius for all sulfate aerosols, both tropospheric and stratospheric. We used the default aerosol size, which is calibrated to represent tropospheric sulfate and the background sulfate layer in the stratosphere but is far too small for the aerosols that would result from a high stratospheric loading of sulfate. Past experiments using an earlier version of this model increased the aerosol size, resulting in a better match to stratospheric sulfate aerosols but was far larger than should have occurred for tropospheric aerosols. This had non-negligible effects on radiative forcing, tropospheric chemistry, and aerosol deposition \citep[e.g.,][]{kravitz2009}, limiting the ability to compare the SAI run with a corresponding baseline run. The approach chosen in the present study avoids these issues, but in doing so limits the applicability of the GISS-OMA simluations to SAI. Nevertheless, these simulations serve as a useful point of comparison and reveal understanding, so they are kept in the manuscript.*"

**Specific comments**

**Section 2.1-to 2.2: I suggest harmonizing the three model descriptions. CESM2 has comprehensive stratospheric chemistry and simplified tropospheric chemistry, what about GISS and UKESM? GISS only mentioned heterogeneous chemistry, UKESM doesn't mention anything at all. I would at least mention if UKCA is bulk, modal, or sectional and if it's coupled to the chemistry. I know they are described better below but all three descriptions should have the same format.**

Thank you for the suggestion. We have tried to homogenize the three sections, adding specific information on the aerosol and chemistry schemes for all models.

**Line 145: ". Condensational growth leading to a transfer between Aitken and Accumulation modes is also treated differently than in the other two models" differently how?**

We have tried to specify this better in the revised version. We now specify

"*To prevent the mean diameter of the Aitken mode from approaching that of the Accumulation mode too often, a transfer function that completely moves all particles from the Aitken to the*

*Accumulation mode when the two sizes approach in value is used (see section 2.8 in Bauer et al. (2008))."*

**Table 1: I imagine that the GISS bulk model also assumes a size distribution, for instance, to calculate the optical properties and that the 0.3 um is the modal radius of the fixed size distribution. Is that the case for OMA (If so, a standard deviation must be specified for the prescribed mode) or does OMA really prescribes that all particles are 0.3 um? Also, I would add the aerosol effective radius that is simulated by the three models with microphysics. Lastly, how many ensemble members have been performed? I don't think I have found it anywhere.**

We have specified the number of ensemble members at the beginning of Section 2. For the effective radius, that is a result that we discuss later on, so we'd rather not include it in this table as it would require too much discussion that is found in following sections.

For OMA, there is a single specified dry radius for sulfate aerosols (0.15 μm), and the model grows those particles consistent with the formulas of Tang (1996) based on the ambient relative humidity (which is less than 20% in the stratosphere) to form a gamma distribution.

**Line 149: I'd specify the diameter here rather than the radius, to avoid confusion with the table where the diameter is specified.**

Thank you for the suggestion, done.

**L165: I am not sure I understand why choosing 22km over 25 km would make it easier to inject in one grid box. Also, it is not clear what "same grid box" refers to. Same across models (I suspect it's not because they have different layers)? Same in time? I am confused by this paragraph.**

The 22/25km issue and the "one gridbox" issue are separate: we have tried to be more clear now. Basically, for the second point, considering that models need to convert the prescribed altitude in pressure level, and select gridboxes based on that, we checked and found out that 22km allowed all models to inject surely in always the same gridbox (since some bodel have hybrid coordinates, the actual equivalence between km of altitude and gridbox may vary seasonally).

**L 188: I do not understand the goal of the second half of this paragraph, starting from Line 186. Is the point to say that the authors don't care about the fact that the same injection leads to very different AOD? I don't agree with including a sentence like this since it is a pretty fundamental conversion that models should agree on. Rather than this, an attempt should be made to explain why here is a difference. Is it because the SO4 removal is less efficient (maybe the particles are smaller in GISS bulk than with explicit microphysics) or because of the different aerosol optical properties due to the different sizes? Is it possible to include the effective radii calculated in all models, to see how they**

**compare with each other and with GISS bulk, as well as the SO4 burden? This is partly answered in Fig. 4, and it would be good to mention it here.**

We have tried to be more precise here: there are various kinds of uncertainties, and what we meant is that if you use a control algorithm, the uncertainty related to more or less SO2 needed to achieve a certain AOD is "moved over" to the decisions of the control algorithm (but doesn't disappear). "Less important" was a very poor choice of words. We have rephrased to:

"The uncertainties in the SAI process that leads from SO2 injection to surface impacts can be separated into three main parts: i) at each latitude how much SO2 is needed to achieve a certain optical depth (i.e. the efficiency of SO2 to H2SO4 conversion and of the removal processes); ii) the resulting distribution of AOD under specified injection location(s) (largely driven by large scale dynamics and mixing) and iii) the impacts of a specific aerosol distribution on climate. Simulations with fixed SO2 injections at fixed locations, as used in this manuscript, allow to explore better points (i) and (ii). In contrast, simulations like those described in (Kravitz et al., 2017), where the amount of SO2 injected is adjusted each year in order to achieve some specified surface temperature goals, allow to better understand the response to some specific pattern of AOD forcing (as the injection rate can be adjusted to achieve a desired AOD). In the following analyses we will thus often separate and discuss both the overall simulated zonal mean response and that normalized by the global magnitude of the response to highlight these different contributors to the overall uncertainty."

**L 210: I imagined the models must have been compared to observations at some point. It would be helpful here to give a description of how each model compared to observations with respect to basic stratospheric circulation: for instance, is UKESM known to have a too**
**isolated tropical pipe or to strong vertical transport in the tropical stratosphere? What about interannual variability: are the simulated variabilities similar to the observed ones (I mean in control simulations that must have been performed in the past).**

As noted above, an evaluation of baseline circulation is the subject of PART2 of this study, where its role in contributing to the simulated aerosol distributions is also discussed in depth. We have taken care to reference the relevant sections of PART2 throughout the current manuscript (Part 1).

**Fig. 3 needs improvement. The labels and ticks of the color scale are illegible. Since the same color scale is applied to all panels, I suggest using one larger color bar at the bottom of the figure, and also enlarging the fonts on the axis.**

We have updated figures 3 and 5 as suggested. Other figures have been improved as well.

**Line 237: I think it's panel 4h, not 4g.**

Fixed, thank you.

**L240: as I mention above, I suspect OMA assumes a lognormal distribution with a modal radius of 0.15 micron. If that's the case, the effective radius can be calculated for OMA using relationships between modal and effective radius in lognormal distributions (I think it's in Seinfeld and Pandis, but in any case is also included in Aquila et al. 2012). If that's the case, I suggest adding the effective radius for OMA for comparison.**

Pretty close - see the response above to the comment on Table 1. It's difficult to use these formulas because there isn't a specified distribution width. Nevertheless, the aerosol size is much smaller than would be expected from an actual SAI deployment. We have downplayed the role of OMA in this manuscript and only use it as a point of comparison.

**L253: number of the supplementary figure is missing**

Fixed, thank you!

**L257 radius _IN_ GISS model**

Added.

**L272: there are three "for instance" in three lines.**

We have modified the phrase.

**L286: how many models were included in the multi-model average of GeoMIP G6?**

Six. We have added this information in the text.

**L297 and following: the discussion about tuning is quite vague and can be made more precise by looking into the model setup and seeing which tuning parameters have been changed to keep remedy the low background AOD. Also, I am not sure I understand the reasoning; the background (non-SAI) AOD can be verified against observations, and comparing against observations could tell us whether 0.03 or 0.11 is more reasonable. If 0.03 is too low (compared to observations) the most obvious "fix" to me seems like increasing emissions, or decreasing the radius, rather than changing the temperature sensitivity to aerosols. Also, which tuning parameter would affect the temperature sensitivity to aerosols specifically?**

There are two standard tuning parameters for GISS that affect the high cloud coverage and net radiative flux at TOA (largely through low clouds). Our intent was not necessarily to comment on which version of the model replicates the radius of the background aerosol. It was more along the lines of the findings of Kiehl (2007) where models with different sensitivities have different aerosol forcings. This is an emergent property of the models, not something that is

tuned.  We hypothesize (with what we believe is good evidence) that the OMA and MATRIX versions of the models have different sensitivities to forcings in general, which would also emerge in terms of different tuning parameter values, and as such have different sensitivities to aerosol forcing (geoengineering).  We have attempted to clarify this in the manuscript.

**Fig. 6: the letters identifying the panels are missing**

**L343: one "at" too many**

We've removed the first "at". Thanks!

**L353: what is the difference between (I0, I1, I2) and (L0, L1, L2)? Generally, I find this explanation a bit confusing. It's pretty clear in Kravitz et al. (2016). I would either make it longer and more explicit, or shorter and more qualitative with an explicit reference to go look in Kravitz et al. (2016). It is a bit difficult to keep in mind the physical meaning of what the text explains. I have also found this section quite disconnected from the previous ones in terms of style and clarity, at the point that it could be moved to a different paper where it would be easier to expand on the meaning of the results.**

Based on this comment from both reviewers, we have modified Section 5 to further clarify all aspects of this portion of the work.

---

## Author Response (AR2)

Dear editor,
thank you for your support and feedback.
We have uploaded a finalized version with all required technical changes as requested by you and the reviewers, and a tracked version of all changes.

Best,
Daniele Visioni on behalf of all authors